

# Effects of environmental hypoxia and hypercarbia on ventilation and gas exchange in Testudines

Pedro Trevizan-Baú[1,2], Augusto S. Abe[3] and Wilfried Klein[1]

[1] Departmento de Biologia, Faculdade de Filosofia, Ciências e Letras de Ribeirão Preto, Universidade de São Paulo, Ribeirão Preto, SP, Brazil
[2] Programa de Pós-graduação em Biologia Comparada, Universidade de São Paulo, Ribeirão Preto, SP, Brazil
[3] Instituto de Biociências, Universidade Estadual Paulista, Rio Claro, SP, Brazil

## ABSTRACT

**Background**. Ventilatory parameters have been investigated in several species of Testudines, but few species have had their ventilatory pattern fully characterized by presenting all variables necessary to understand changes in breathing pattern seen under varying environmental conditions.

**Methods**. We measured ventilation and gas exchange at 25 °C in the semi-aquatic turtle *Trachemys scripta* and the terrestrial tortoise *Chelonoidis carbonarius* under normoxia, hypoxia, and hypercarbia and furthermore compiled respiratory data of testudine species from the literature to analyze the relative changes in each variable.

**Results**. During normoxia both species studied showed an episodic breathing pattern with two to three breaths per episode, but the non-ventilatory periods ($T_{NVP}$) were three to four times longer in *T. scripta* than in *C. carbonarius*. Hypoxia and hypercarbia significantly increased ventilation in both species and decreased $T_{NVP}$ and oxygen consumption in *T. scripta* but not in *C. carbonarius*.

**Discussion**. Contrary to expectations, the breathing pattern in *C. carbonarius* did show considerable non-ventilatory periods with more than one breath per breathing episode, and the breathing pattern in *T. scripta* was found to diverge significantly from predictions based on mechanical analyses of the respiratory system. A quantitative analysis of the literature showed that relative changes in the ventilatory patterns of chelonians in response to hypoxia and hyperbarbia were qualitatively similar among species, although there were variations in the magnitude of change.

## INTRODUCTION

The order Testudines differs from the other groups of reptiles by the presence of a rigid shell, impeding lung ventilation through movement of the ribs (*Lyson et al., 2014*). To overcome this morphological constraint, Testudines contract abdominal muscles associated with their legs, thereby compressing or expanding the body cavity and resulting in lung ventilation (*Gans & Hughes, 1967*; *Gaunt & Gans, 1969*).

Testudines can be divided into two suborders. The 100 species of Pleurodira are characterized by a retraction of the neck in the horizontal plane, whereas the 250 species

Corresponding author
Wilfried Klein, wklein@usp.br

of Cryptodira retract their neck in the vertical plane (*Werneburg et al., 2015*; *Uetz, Freed & Hošek, 2018*). All Pleurodira occur in freshwater habitats, just as the majority of cryptodiran species. However, some Cryptodira live in the marine environment and all representatives of the family Testudinidae and some species of Emydidae are terrestrial.

Gas exchange and ventilation have been studied in several species of turtles, tortoises, and terrapins, but few species were fully characterized regarding their breathing variables. In particular, two species of the family Emydidae (Cryptodira), the semi-aquatic *Trachemys scripta* and *Chrysemys picta* have been used in numerous respiratory studies (see Table S1). Although oxygen consumption has been determined in many chelonian species (see *Ultsch, 2013*, for review), data on ventilatory parameters, such as overall breathing frequency, tidal volume, and minute ventilation, are available only for a small number of species, representing a limited range of the taxonomic diversity, especially when considering responses to hypoxic and hypercarbic exposures (Table S1). While the number of studies listed in Table S1 seems extensive, few have actually characterized the ventilatory pattern by providing data such as inspiratory time, expiratory time, total duration of a ventilatory cycle, duration of the non-ventilatory period, breathing frequency during breathing episodes, frequency of breathing episodes, as well as breathing frequency, tidal volume, and minute ventilation (*Benchetrit & Dejours, 1980*; *Cordeiro, Abe & Klein, 2016*). Furthermore, most of these data have been obtained for *Chrysemys picta* (e.g., *Milsom & Jones, 1980*; *Milsom & Chan, 1986*; *Funk & Milsom, 1987*; *Wasser & Jackson, 1988*). The totality of these variables is needed to fully characterize the ventilatory behavior of a species under varying environmental conditions, especially in ectothermic vertebrates where ventilation can show highly episodic burst breathing or regular singlet breathing (for review see *Shelton, Jones & Milsom, 1986*). *Fong, Zimmer & Milsom (2009)* suggested that an increasing respiratory drive changes episodic into continuous breathing and *Johnson, Krisp & Bartman (2015)* used duration of inspiration and expiration as measures for inspiratory and expiratory drive, respectively. Furthermore, *Milsom & Wang (2017)* argued that the regulation of the ventilatory responses is complex and cannot be totally understood with few variables measured, especially because Testudines possess an undivided heart that allows for intracardiac shunting of blood between the pulmonary and systemic circulations.

Among the Testudines, the terrestrial species belonging to the family Testudinidae are also very poorly characterized regarding their ventilatory response to hypoxia or hypercarbia, and only data on breathing frequency, tidal volume, minute ventilation, and oxygen consumption are available (*Altland & Parker, 1955*; *Benchetrit, Armand & Dejours, 1977*; *Benchetrit & Dejours, 1980*; *Burggren, Glass & Johansen, 1977*; *Glass, Burggren & Johansen, 1978*; *Ultsch & Anderson, 1988*). *Burggren, Glass & Johansen (1977)* and *Glass, Burggren & Johansen (1978)* showed that under normoxic conditions, the terrestrial *Testudo pardalis* employs regular single breaths separated by short breath-holds. A regular singlet breathing behavior has also been shown by *Burggren (1975)* for the tortoise *Testudo graeca* and by *Benchetrit, Armand & Dejours (1977)* for *Testudo horsfieldi*. The semi-aquatic *Pelomedusa subrufa*, on the other hand, uses breathing episodes containing several ventilations interspaced by longer breath-holds (*Burggren, Glass & Johansen, 1977*; *Glass, Burggren & Johansen, 1978*). Such a pattern has been interpreted as adaptation to the

aquatic life-style observed in *P. subrufa* and other aquatic or semi-aquatic species, where the episodic breathing reduces the amount of time spent at the water surface, reducing the risk of predation, as well as reducing the cost of ascending to the surface (*Randall et al., 1981*).

Depending on the gas concentration, hypoxia, as well as hypercarbia, stimulates breathing in turtles, with moderate concentrations of hypercarbia generally increasing ventilation more than very low oxygen concentrations (*Shelton, Jones & Milsom, 1986*). Interestingly, *Altland & Parker (1955)* found a more episodic breathing pattern in *Terrapene carolina carolina* under normoxia that changed to a more regular singlet breathing pattern under hypoxic conditions. The normal response to either hypoxia or hypercarbia results in reduced non-ventilatory periods, but may or may not increase breathing frequency or tidal volume (*Shelton, Jones & Milsom, 1986*). In a recent study, *Cordeiro, Abe & Klein (2016)* demonstrated that two closely related pleurodirans exhibit different ventilatory responses to hypoxia and hypercarbia. While both species reduce significantly the non-ventilatory period and increase breathing frequency during hypoxic and hypercarbic exposures, *Podocnemis unifilis* significantly increases the breathing frequency during breathing episodes during hypercarbia but significantly decreases the breathing frequency during breathing episodes during hypoxia, whereas *Phrynops geoffroanus* significantly decreases breathing frequency during breathing episodes under hypoxia, but does not change this variable during hypercarbia.

Given these variations in the breathing pattern during normoxia, hypoxia, and hypercarbia among Testudines, and considering the very few ventilatory data available for terrestrial species, the aim of the present study was to analyze the ventilatory response to different gas mixtures in two cryptodirans, the red-eared slider *Trachemys scripta* (Emydidae) and the South American red-footed tortoise *Chelonoidis carbonarius* (Testudinidae). *Trachemys scripta*, the model species for cardiorespiratory studies, was investigated because no previous study reported all ventilatory variables obtained from the same animals and experimental protocols, both under hypoxic and hypercarbic conditions, whereas *C. carbonarius* was chosen because it is a widespread South American tortoise that has not had its respiratory physiology investigated previously. Furthermore, the present data were compiled together with available data from the literature to characterize the general response of Testudines to hypoxia and hypercarbia and to verify if terrestrial species show a significantly different ventilatory pattern compared to semi-aquatic species.

## MATERIALS AND METHODS
### Animals
Adults of both sexes of *T. scripta* (body mass: $M_B = 1.08 \pm 0.10$ kg; $N = 8$) and *C. carbonarius* ($M_B = 3.77 \pm 0.61$ kg; $N = 6$) living under natural conditions were obtained from the Jacarezário, Univeridade Estadual Paulista "Júlio de Mesquita Filho", Rio Claro, SP, Brazil, transported to the laboratory at the University of São Paulo in Ribeirão Preto, SP, and maintained for at least three months before experimentation to acclimate to laboratory conditions. Experiments were performed between November 2014

and February 2015 following approval by the Instituto Chico Mendes de Conservação da Biodiversidade (SISBIO; license number 35221-1) and Comissão de Ética no Uso de Animais (CEUA USP/Campus de Ribeirão Preto; protocol number 12.1.1541.53.0). Animals were maintained under a 12 h light/dark photoperiod cycle, in a temperature-controlled room at 25 ± 2 °C and received a mixed diet supplemented with amino acids, vitamins and minerals (Aminomix Pet, Vetnil®, Louveira, Brazil) three times a week. *T. scripta* were housed in a box with a water reservoir for diving whereas *C. carbonarius* were housed in boxes whose bottom was covered with wooden chips.

## Setup

Animals were submitted to open respirometry following *Glass, Wood & Johansen (1978)*, *Wang & Warburton (1995)* and *Silva et al. (2011)* to measure ventilation and gas exchange. Individuals of *T. scripta* were placed in an aquarium with a single access to an inverted funnel, and each individual only needed to extend its neck and protrude its nostrils into the chamber for air breathing. *C. carbonarius* were placed in a plastic box and a mask was fitted to the head of each animal for respirometry and a collar was fixed to the neck to prevent head retraction. The dead space of the funnel or the mask was never larger than 40 ml. The exit of the funnel and the frontal tip of the mask were equipped with a pneumotach (Fleisch tube), which was connected to a spirometer (FE141, ADInstruments, Sydney, Australia). The gas inside the funnel or mask was sampled at 180 ml min$^{-1}$, dried, and pulled to a gas analyzer (ML206, ADInstruments, Sydney, Australia). Data were recorded and analyzed using PowerLab 8/35 and LabChart 7.0 (ADInstruments, Sydney, Australia).

Both the funnel and the mask were calibrated by injections of known volumes, using an Inspira ventilator (Harvard Apparatus, Cambridge, MA, USA), and concentrations of gas, supplied by a Pegas 4000MF gas mixer (Columbus Instruments Columbus, OH, USA). Air was used for the spirometer calibration with volumes ranging from 1 to 60 ml and different volumes and concentrations of $O_2$ were used to calibrate the gas exchange measurements. In all cases, calibrations resulted in linear regressions with $R^2$ >0.95.

## Experimental protocols

Experimental temperature and photoperiod were the same as during maintenance, and all animals were fasted for three to seven days before experimentation to avoid the confounding effects of digestion on metabolism. The animals were weighed one day before the beginning of each experimental treatment. Before any measurements, animals were placed into the experimental setup or equipped with a mask at least 12 h before initiation of experiments. Experimentation started around 08:00, and ventilation and gas exchange were measured first under normoxic conditions, followed by progressively decreasing hypoxic (9, 7, 5, 3% $O_2$) or progressively increasing hypercarbic (1.5, 3.0, 4.5, 6.0% $CO_2$) exposures, and after that animals were exposed again to normoxia. The exposure time of each gas mixture, as well as normoxic conditions, was 2 h. Animals were first exposed to one randomly chosen progressive gas exposure and the following day to the other one.

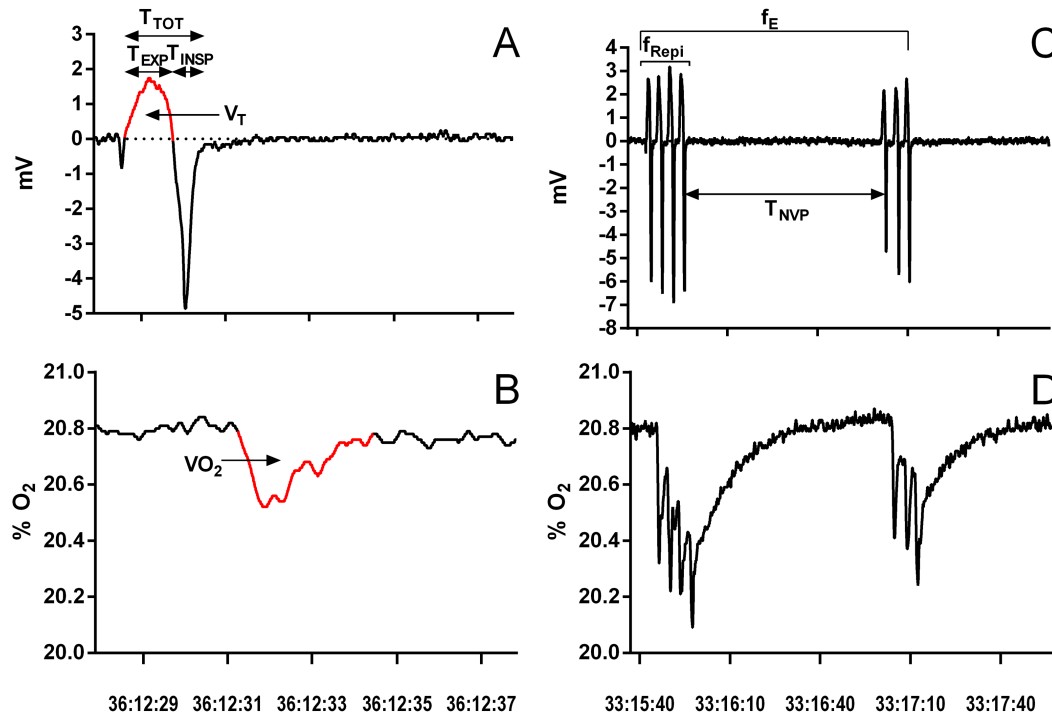

**Figure 1** **Example of a single ventilatory cycle and episodic ventilation in *Chelonoidis carbonarius* during normoxia showing respiratory variables measured.** Traces of a single ventilatory cycle (A, B) and episodic ventilation (C, D) in *Chelonoidis carbonarius* during normoxia showing respiratory variables measured A, C: ventilation; B, D: oxygen consumption. For abbreviations see 'Material and Methods'. The red parts on parts A and B represent the areas integrated to determine tidal volume and oxygen consumption, respectively.

## Data analysis

The last hour of each exposure was used to extract the following data: breathing frequency during breathing episodes ($f_{Repi}$, breaths episode$^{-1}$), frequency of breathing episodes ($f_E$, episodes.h$^{-1}$), duration of non-ventilatory period ($T_{NVP}$, s; defined as the time between the end of an inspiration and the beginning of the following expiration), duration of inspiration ($T_{INSP}$, s), duration of expiration ($T_{EXP}$, s), total duration of one ventilatory cycle ($T_{TOT} = T_{EXP} + T_{INSP}$, s), tidal volume ($V_T$, ml kg$^{-1}$), breathing frequency ($f_R$, breaths min$^{-1}$), and oxygen consumption ($VO_2$, mlO$_2$ kg$^{-1}$) (Fig. 1). During episodic breathing, due to the slow response time of the oxygen analyzer, oxygen consumption was determined by integrating the area above the oxygen trace for the entire episode, and then divided by the number of expirations to obtain mean oxygen consumption per breath. From the extracted data, the instantaneous breathing frequency (f′, breaths min$^{-1}$), the relative duration of expiration ($T_{EXP}/T_{TOT}$), the relation between inspiration and expiration ($T_{INSP}/T_{EXP}$), the expiratory flow rate ($V_T/T_{EXP}$, ml s$^{-1}$), minute ventilation ($\dot{V}_E$, ml kg$^{-1}$ min$^{-1}$), oxygen consumption ($\dot{V}O_2$, mlO$_2$ kg$^{-1}$ min$^{-1}$), and air convection requirement ($\dot{V}_E/\dot{V}O_2$, ml mlO$_2^{-1}$) were calculated.

Data were analyzed using GraphPad Prism 6.0 and applying Repeated Measures ANOVA followed by a Tukey's multiple comparison test. Values of $P < 0.05$ were considered significant.

To compare the results of the present study with previously published data, we searched for relevant publications using Pubmed, Web of Science, and Google Scholar databases using keywords such as 'turtle', 'Testudines', 'hypoxia', 'hypercarbia', 'hypercapnia', 'ventilation', 'gas exchange', etc. Values of respiratory variables of Testudines obtained under exposure to environmental hypoxia or hypercarbia (but not anoxia or hypoxic-hypercarbia) measured at temperatures between 20 and 30 °C were included (Table 1). Data from animals that had their trachea cannulated, or that had their respiratory system surgically manipulated, were not included. Values were directly obtained from the text or tables given, or by extracting values from published figures using the free software PlotDigitizer (version 2.6.2). To enable comparison among species, data were expressed as changes relative to normoxic values. Due to the very low number of chelonian species with a complete set of respiratory variables available and the varying experimental protocols applied at different temperatures, levels of hypoxia or hypercarbia, phylogenetically informed multivariate analysis was not possible.

## RESULTS

### Ventilation and oxygen consumption in *T. scripta* and *C. carbonarius*

During normoxia, both species showed an episodic breathing pattern with two to three ventilatory cycles interspersed by non-ventilatory periods (Fig. 2). The $T_{NVP}$ was, on average, three to four times longer in *T. scripta* than in *C. carbonarius*. In *T. scripta*, hypoxia significantly increased $f_E$, $V_T$, $\dot{V}_E$ and $\dot{V}_E/\dot{V}O_2$, whereas $f_{Repi}$, $T_{NVP}$ and $\dot{V}O_2$ were significantly reduced (Figs. 3–6). Under hypoxic exposure, *C. carbonarius* significantly increased $f_E$, $T_{INSP}$, $V_T$, $f_R$, $\dot{V}_E$, and $\dot{V}_E/\dot{V}O_2$ (Figs. 3–6). Once the hypoxic exposure ended, all variables returned to pre-hypoxic values within 1 hour, with the exception of $f_R$ in *C. carbonarius*, which was significantly greater when compared to the pre-hypoxic value. Exposure to $CO_2$ increased $\dot{V}_E$ and $\dot{V}_E/\dot{V}O_2$ significantly and decreased $\dot{V}O_2$ significantly in *T. scripta*, whereas in *C. carbonarius* $T_{INSP}$, $T_{TOT}$, $V_T$, $f_R$, $\dot{V}_E$, and $\dot{V}_E/\dot{V}O_2$ significantly increased but $T_{NVP}$ and f' significantly decreased (Figs. 3–6). One hour after the withdrawal of $CO_2$, all variables had returned to pre-hypercarbic values. The relationships between $T_{EXP}$, $T_{INSP}$ and $T_{TOT}$ (i.e., $T_{EXP}/T_{TOT}$, and $T_{INSP}/T_{EXP}$ respectively) were not significantly affected by either hypoxia nor hypercarbia, just as expiratory flow rate ($V_T/T_{EXP}$), but the latter did show a tendency to increase in both species with increasing levels of hypoxia and hypercarbia (Fig. 5).

### Relative changes in respiratory variables

Both hypoxia and hypercarbia increased ventilation. This increase was achieved by increasing the number of breathing episodes, caused by decreasing the non-ventilatory period (Fig. 7). $T_{NVP}$ at 3% $O_2$, for example, consistently represented about 20% of the $T_{NVP}$ seen during normoxia in all species investigated, whereas 6% $CO_2$ roughly reduced $T_{NVP}$ by 50%. Interestingly, hypercarbia about doubled $f_E$, with the exception of *P. geoffroanus*,

**Table 1  Respiratory variables extracted from the literature.**

| Species | \multicolumn{12}{c}{Respiratory variables} | Reference |
|---|---|---|---|---|---|---|---|---|---|---|---|---|---|
| | $f_{Repi}$ | $f_E$ | $T_{NVP}$ | $T_{INSP}$ | $T_{EXP}$ | $T_{TOT}$ | $f'$ | $V_T$ | $f_R$ | $\dot{V}_E$ | $\dot{V}O_2$ | $\dot{V}_E/\dot{V}O_2$ | |
| **Hypoxia** | | | | | | | | | | | | | |
| *Chelonoidis carbonarius* | x | x | x | x | x | x | x | x | x | x | x | x | This study |
| *Chelydra serpentina* | | x | | | | | | | | | | | *Boyer (1963)* |
| *Chelydra serpentina* | | | | | | | | | | | x | | *Boyer (1966)* |
| *Chelydra serpentina* | | | | | | | | x | x | x | | | *Frische, Fago & Altimiras (2000)* |
| *Chelydra serpentina* | x | | x | | | | | | | | | | *West, Smits & Burggren (1989)* |
| *Chrysemys picta* | | x | | | | | | x | x | x | x | x | *Glass, Boutilier & Heisler (1983)* |
| *Chrysemys picta* | x | x | x | | | | | x | x | x | | | *Milsom & Chan (1986)* |
| *Gopherus polyphemus* | | | | | | | | | | | x | | *Ultsch & Anderson (1988)* |
| *Pelomedusa subrufa* | | | | | | | | x | x | x | | | *Burggren, Glass & Johansen (1977)* |
| *Pelomedusa subrufa* | | x | | | | | | x | x | x | | | *Glass, Burggren & Johansen (1978)* |
| *Phrynops geoffroanus* | x | x | x | x | x | x | x | x | x | x | x | x | *Cordeiro, Abe & Klein (2016)* |
| *Podocnemis unifilis* | x | x | x | x | x | x | x | x | x | x | x | x | *Cordeiro, Abe & Klein (2016)* |
| *Terrapene carolina* | | | | | | | | x | x | x | x | | *Altland & Parker (1955)* |
| *Terrapene carolina* | | | | | | | | | | | x | | *Ultsch & Anderson (1988)* |
| *Testudo horsfieldi* | | | | | | | | | | x | | | *Benchetrit, Armand & Dejours (1977)* |
| *Testudo pardalis* | | | | | | | | x | x | x | | | *Burggren, Glass & Johansen (1977)* |
| *Testudo pardalis* | | x | | | | | | x | x | x | | | *Glass, Burggren & Johansen (1978)* |
| *Trachemys scripta* | x | | | | | x | | | | | | | *Frankel et al. (1969)* |
| *Trachemys scripta* | | | | | | | | | | | x | | *Hicks & Wang (1999)* |
| *Trachemys scripta* | | | | | | | | | | | x | | *Jackson & Schmidt-Nielsen (1966)* |
| *Trachemys scripta* | x | x | | | | | | x | x | x | x | x | *Lee & Milsom (2016)* |
| *Trachemys scripta* | x | x | x | x | x | x | x | x | x | x | x | x | This study |
| *Trachemys scripta* | | | | | | x | x | x | x | x | | | *Vitalis & Milsom (1986b)* |
| **Hypercarbia** | | | | | | | | | | | | | |
| *Chelonia mydas* | | | | | | | | x | x | x | | | *Jackson, Kraus & Prange (1979)* |
| *Chelonoidis carbonarius* | x | x | x | x | x | x | x | x | x | x | x | x | This study |
| *Chelydra serpentina* | x | | x | | | | | | | | | | *West, Smits & Burggren (1989)* |
| *Chrysemys picta* | x | x | x | | | | | x | x | x | x | x | *Funk & Milsom (1987)* |
| *Chrysemys picta* | x | x | x | | | | | x | x | x | | | *Milsom & Chan (1986)* |
| *Chrysemys picta* | x | x | x | x | x | x | | x | x | | | | *Milsom & Jones (1980)* |
| *Chrysemys picta* | x | | x | | | | | x | x | x | x | x | *Silver & Jackson (1985)* |
| *Pelomedusa subrufa* | | | | | | | | x | x | x | | | *Burggren, Glass & Johansen (1977)* |
| *Pelomedusa subrufa* | | x | | | | | | x | x | x | | | *Glass, Burggren & Johansen (1978)* |
| *Phrynops geoffroanus* | x | x | x | x | x | x | x | x | x | x | x | x | *Cordeiro, Abe & Klein (2016)* |
| *Podocnemis unifilis* | x | x | x | x | x | x | x | x | x | x | x | x | *Cordeiro, Abe & Klein (2016)* |
| *Testudo pardalis* | | x | | | | | | x | x | x | | | *Glass, Burggren & Johansen (1978)* |
| *Testudo horsfieldi* | | x | x | x | x | | | x | x | x | | | *Benchetrit & Dejours (1980)* |
| *Testudo pardalis* | | | | | | | | x | x | x | | | *Burggren, Glass & Johansen (1977)* |

| Species | Respiratory variables | | | | | | | | | | | | Reference |
|---------|---------|-------|----------|----------|---------|----------|------|-------|-------|-----------|-----------|---------------|-----------|
| | $f_{Repi}$ | $f_E$ | $T_{NVP}$ | $T_{INSP}$ | $T_{EXP}$ | $T_{TOT}$ | $f'$ | $V_T$ | $f_R$ | $\dot{V}_E$ | $\dot{V}O_2$ | $\dot{V}_E/\dot{V}O_2$ | |
| *Trachemys scripta* | x | | | | | x | | | | | | | *Frankel et al. (1969)* |
| *Trachemys scripta* | | | | | | | | | | | x | | *Hicks & Wang (1999)* |
| *Trachemys scripta* | | | | | | | | x | x | x | | | *Hitzig & Nattie (1982)* |
| *Trachemys scripta* | | | | | | | | x | x | x | x | x | *Jackson, Palmer & Meadow (1974)* |
| *Trachemys scripta* | x | | | | | | | x | x | x | | | *Johnson & Creighton (2005)* |
| *Trachemys scripta* | x | x | x | x | x | x | x | x | x | x | x | x | This study |
| *Trachemys scripta* | | | | | | x | x | x | x | x | | | *Vitalis & Milsom (1986b)* |

**Notes.**

Abbreviations: $f_{Repi}$, breathing frequency during breathing episodes; $f_E$, number of breathing episodes; $T_{NVP}$, duration of non-ventilatory period; $T_{INSP}$, duration of inspiration; $T_{EXP}$, duration of expiration; $T_{TOT}$, total duration of one ventilatory cycle; $f'$, instantaneous breathing frequency; $V_T$, tidal volume; $f_R$, breathing frequency; $\dot{V}_E$, minute ventilation; $\dot{V}O_2$, oxygen consumption; $\dot{V}_E/\dot{V}O_2$, air convection requirement.

and slightly increased $f_{Repi}$ (exceptions *P. geoffroanus* and *C. carbonarius*), whereas hypoxia caused a greater increase in $f_E$, but slightly decreased $f_{Repi}$ (exception *C. carbonarius*). Neither hypoxia nor hypercarbia drastically altered $T_{INSP}$, $T_{EXP}$, $T_{TOT}$, and $f'$ (Fig. 8), as well as $T_{EXT}/T_{TOT}$ and $T_{INSP}/T_{EXP}$ (Fig. 9).

$V_T/T_{EXP}$ increased two to five-fold under hypoxic and hypercarbic conditions (Fig. 9) in all species studied, which was mainly caused by an about two to three-fold increase in $V_T$ at severe levels of hypoxia and hypercarbia (Fig. 10). *C. carbonarius*, showing a 12-fold, and *P. geoffroanus*, showing a six-fold increase in $V_T$, were the only species showing much larger increases in $V_T$. Several species increased $f_R$ during hypercarbia about six to seven-fold, but many species only doubled or tripled $f_R$ (Fig. 10). The only species that increased $f_R$ more than three-fold during hypoxia were *C. picta* at 30 °C (*Glass, Boutilier & Heisler, 1983*) and *P. geoffroanus* at 25 °C (*Cordeiro, Abe & Klein, 2016*). The product of $V_T$ and $f_R$, minute ventilation, showed the greatest relative increases, with *P. geoffroanus* increasing $\dot{V}_E$ 42 times and *C. carbonarius* about 30 times, both at 6% $CO_2$. The relative increase at 6% $CO_2$ ranged from four to 12 times, whereas at 3% $O_2$ the increase in $\dot{V}_E$ ranged between three and six or between 12 and 17 for *C. picta*, *C. carbonarius* and *P. geoffroanus*.

With the exception of *P. geoffroanus*, both under hypoxia and hypercarbia, and of *P. unifilis* under hypercarbia, $\dot{V}O_2$ decreased or remained unaltered during both exposures (Fig. 11). The resulting air convection requirement, however, increased about 10 to 30-fold in *T. scripta* (*Jackson (1973)* and *Lee & Milsom (2016)* *versus* this study, respectively), in *C. picta* (3% $O_2$; *Glass, Boutilier & Heisler, 1983*), and in *C. carbonarius* (4.5 and 6% $CO_2$; this study) (Fig. 11). In the remaining species, $\dot{V}_E/\dot{V}O_2$ increased about three to 12 times under both hypoxic and hypercarbic conditions.

## DISCUSSION

### Ventilation and oxygen consumption in *T. scripta* and *C. carbonarius*

Breathing pattern of both species followed the general reptilian behavior of intermittent lung ventilation. *Burggren (1975)* and *Glass, Burggren & Johansen (1978)* observed intermittent ventilation in *Testudo graeca* and *T. pardalis*, respectively, but in both species breathing pattern consisted of just one ventilatory cycle interspersed by short and

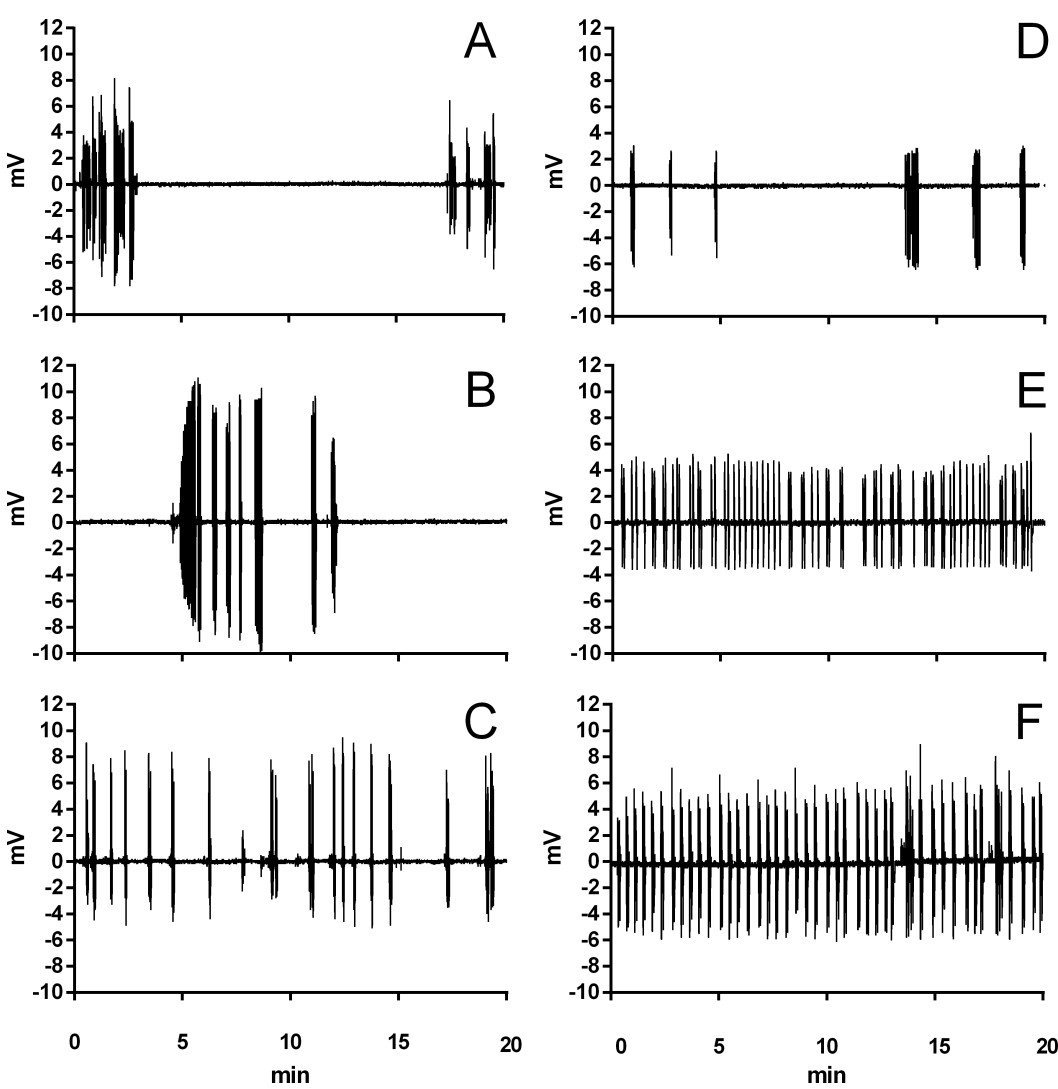

**Figure 2  Example traces of ventilation in *Trachemys scripta* and *Chelonoidis carbonarius*.** Traces of ventilation in *Trachemys scripta* (A–C) and *Chelonoidis carbonarius* (D–F) during normoxia (A, D), 6% $CO_2$ (B, E), and 3% $O_2$ (C, F).

regular non-ventilatory periods. In the present study, both, *T. scripta* and, unexpectedly, *C. carbonarius*, showed more than one ventilatory cycle per breathing episode, but the mean duration of the non-ventilatory periods was lower in *C. carbonarius* when compared to *T. scripta*. *Vitalis & Milsom (1986a)* consider episodic breathing an adaptive mechanism that decreases the energetic cost of ventilation in ectotherms, and *Randall et al. (1981)* consider such a breathing behavior advantageous for aquatic species, since it reduces the energetic cost to surface and also reduces the exposure time at the surface, possibly lessening risks of predation. Since episodic breathing with long non-ventilatory periods leads to a significant change in arterial blood gases, decreasing $P_aO_2$ and pH and increasing $P_aCO_2$ (*Glass, Burggren & Johansen, 1978*), as well as decreasing the efficiency of pulmonary $CO_2$

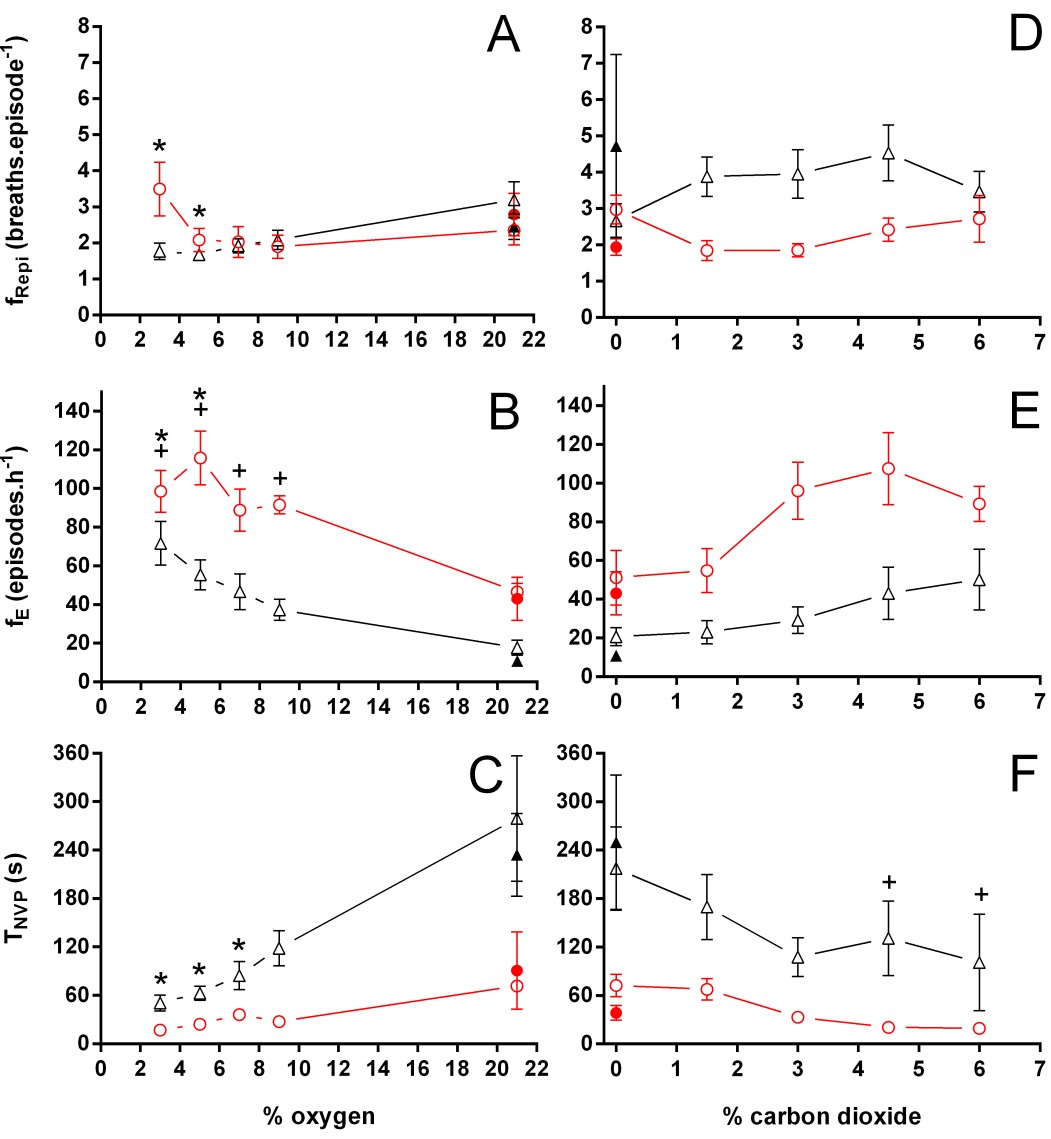

**Figure 3** **Breathing frequency during breathing episodes, number of breathing episodes, and duration of the non-ventilatory period in *Trachemys scripta* and *Chelonoidis carbonarius*.** Breathing frequency during breathing episodes (A, D), number of breathing episodes (B, E), and duration of the non-ventilatory period (C, F) in *Trachemys scripta* (triangle) and *Chelonoidis carbonarius* (circle) during normoxia, hypoxia (A–C) and hypercarbia (D–F) (open symbols) and 1 hour after exposure to the different gas mixtures (closed symbols). * (*T. scripta*) and + (*C. carbonarius*) indicate values significantly different from initial normoxic values

excretion (*Malte, Malte & Wang, 2013*), it should be more advantageous for a terrestrial species to ventilate regularly and thereby maintain homeostasis of arterial blood gases. It is therefore interesting to ask why the terrestrial *C. carbonarius* employs episodic breathing under normoxic conditions, thereby possibly increasing variation in arterial blood gases instead of maintaining a regular breathing pattern, such as seen in this species only under severe levels of hypoxia or hypercarbia (Fig. 2). *C. carbonarius* does frequently

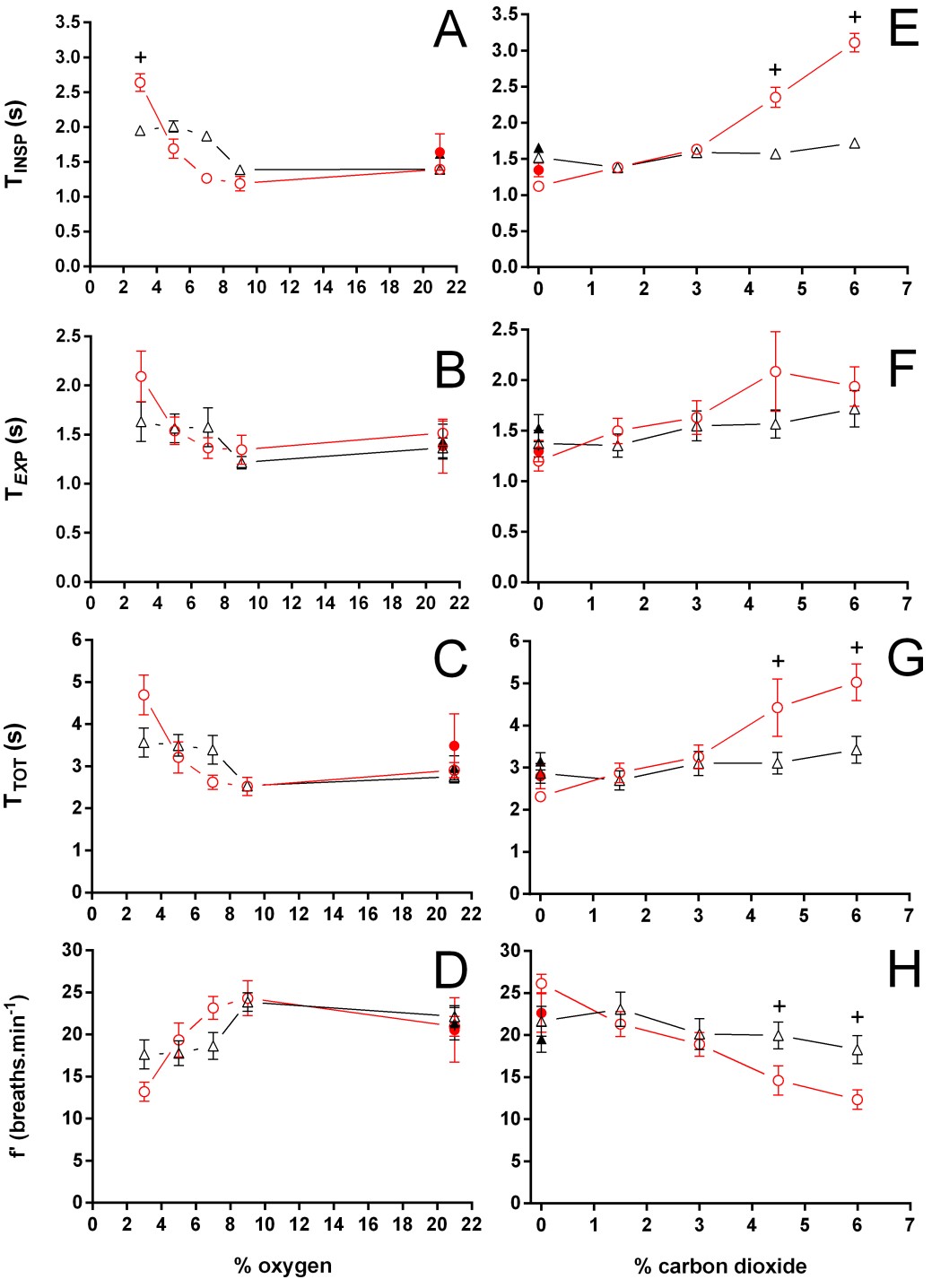

**Figure 4  Duration of inspiration, duration of expiration, total duration of one ventilatory cycle, and instantaneous breathing frequency in *Trachemys scripta* and *Chelonoidis carbonarius*.** Duration of inspiration (A, E), duration of expiration (B, F), total duration of one ventilatory cycle (C, G), and instantaneous breathing frequency (D, H) in *Trachemys scripta* (triangle) and *Chelonoidis carbonarius* (circle) during normoxia, hypoxia (A–D) and hypercarbia (E–H) (open symbols) and 1 hour after exposure to the different gas mixtures (closed symbols). * (*T. scripta*) and + (*C. carbonarius*) indicate values significantly different from initial normoxic values.

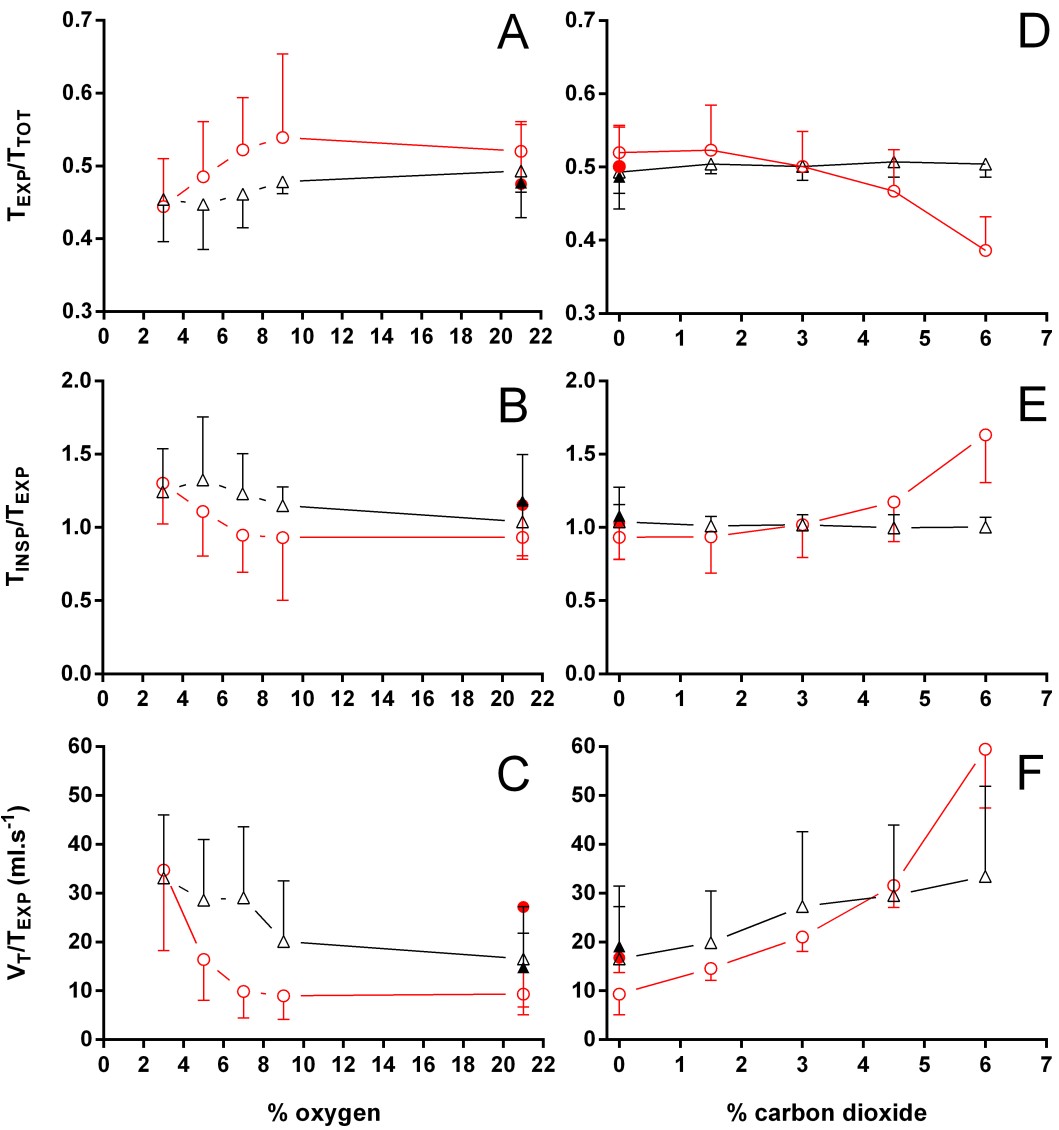

**Figure 5** **Relation between expiration and total duration of one ventilatory cycle, the relation between inspiration and expiration, and the expiratory flow rate in *Trachemys scripta* and *Chelonoidis carbonarius*.** The relation between expiration and total duration of one ventilatory cycle (A, D), the relation between inspiration and expiration (B, E), and the expiratory flow rate (C, F) in *Trachemys scripta* (triangle) and *Chelonoidis carbonarius* (circle) during normoxia, hypoxia (A–C) and hypercarbia (D–F) (open symbols) and 1 hour after exposure to the different gas mixtures (closed symbols).

seek shelter in shallow burrows or other small spaces and remains non-ventilatory for long periods (AS Abe, pers. obs., 2000), possibly explaining the episodic breathing seen in this terrestrial species, but currently a physiological explication for this behavior is lacking. Interestingly, other ectothermic terrestrial species such as varanid (*Thompson & Withers, 1997*) and agamid lizards (*Frappell & Daniels, 1991*) also breathe intermittently. However concomitant blood gas analyses have not been performed in these species to verify accompanying variations in blood gases or pH.

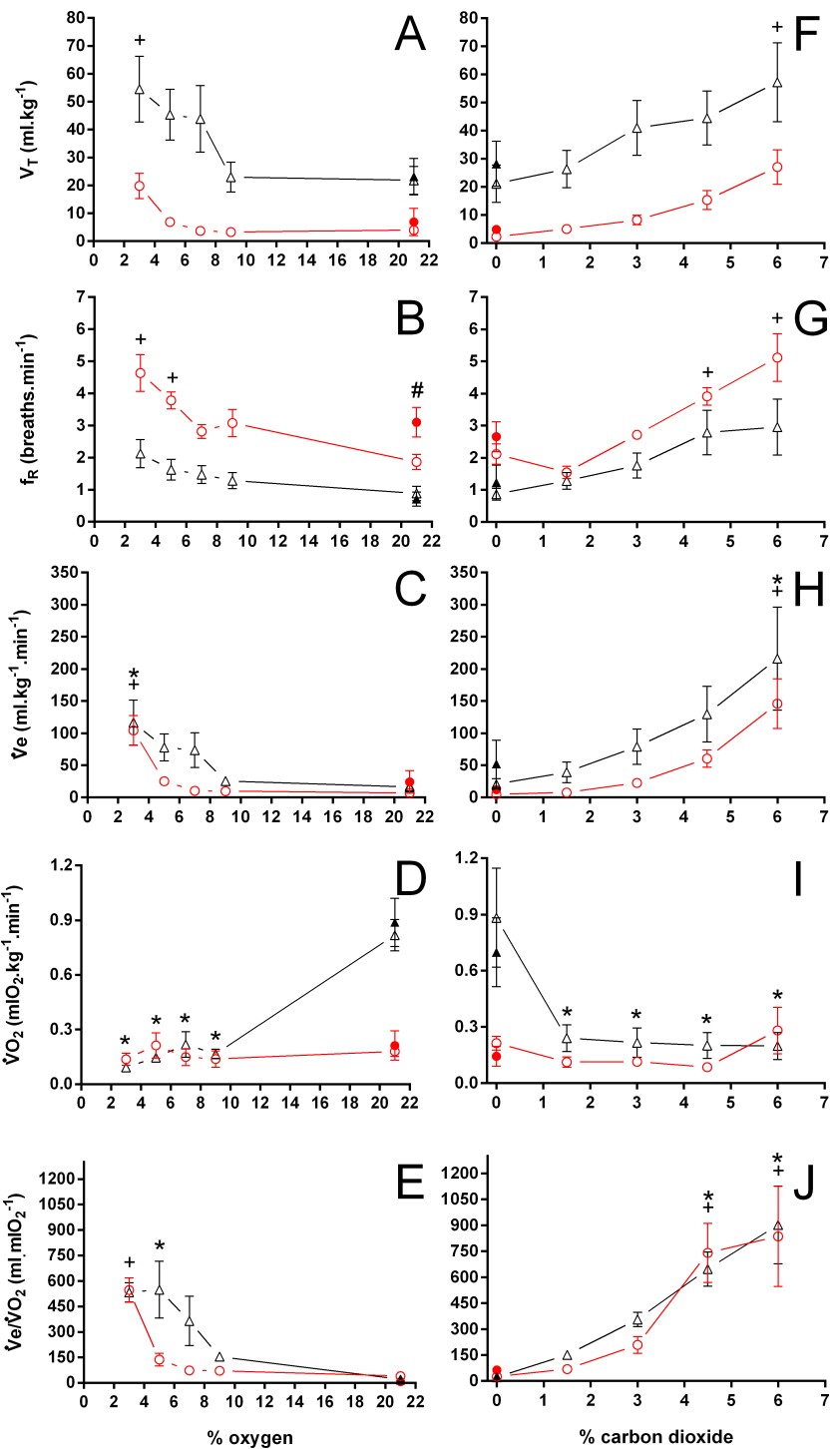

**Figure 6** **Tidal volume, breathing frequency, minute ventilation, oxygen consumption, and air convection requirement in *Trachemys scripta* and *Chelonoidis carbonarius*** Tidal volume (A, F), breathing frequency (B, G), minute ventilation (C, H), oxygen consumption (D, I), and air convection requirement (E, J) in *Trachemys scripta* (triangle) and *Chelonoidis carbonarius* (circle) during normoxia, hypoxia (A–E) and hypercarbia (F–J) (open symbols) and 1 hour after exposure to the different gas mixtures (closed symbols). * (*T. scripta*) and + (*C. carbonarius*) indicate values significantly different from initial normoxic values. # denotes a post-hypoxia normoxic value significantly different from the initial normoxia.

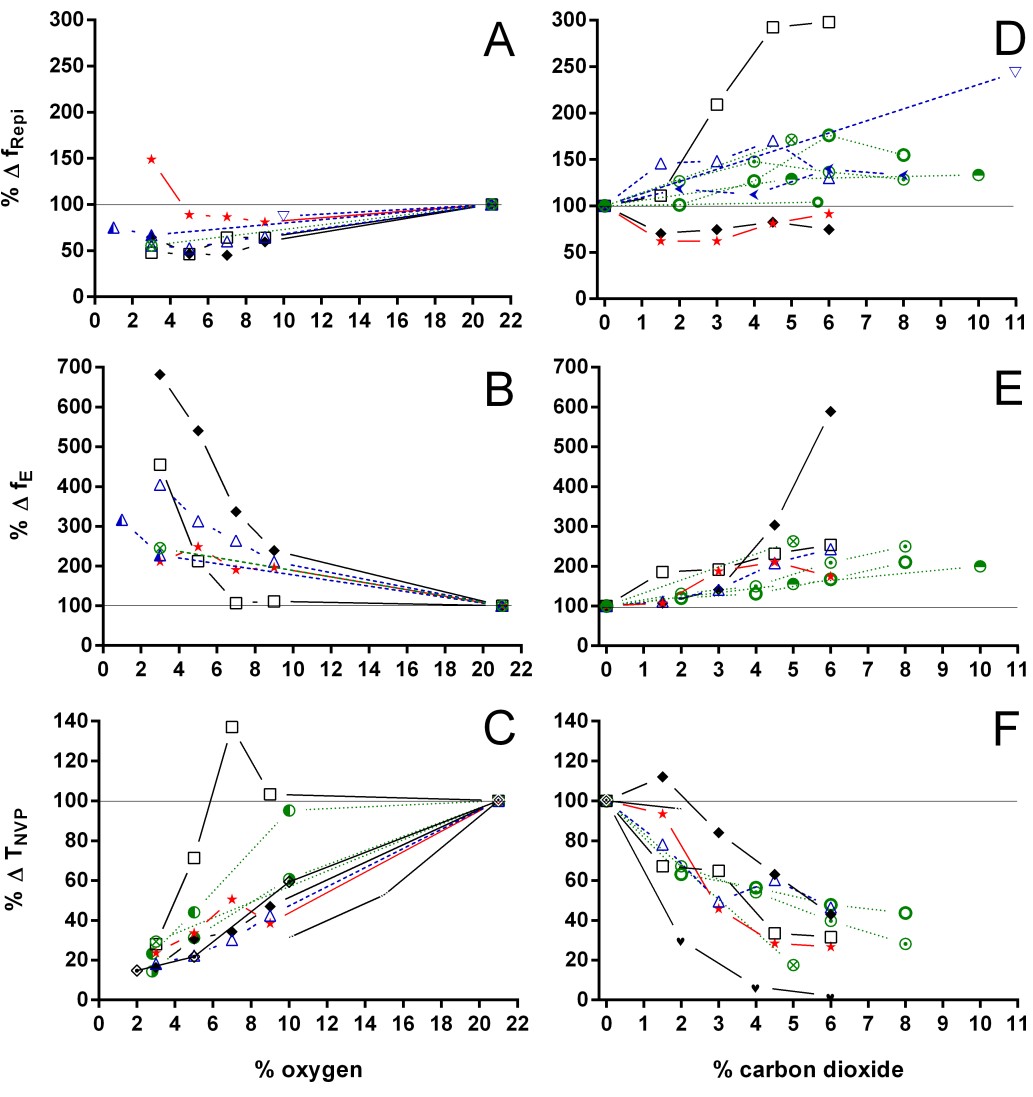

**Figure 7** Relative changes in breathing frequency during breathing episodes, number of breathing episodes, and duration of the non-ventilatory period in Testudines under hypoxic and hypercarbic exposures. Relative changes in breathing frequency during breathing episodes (A, D), number of breathing episodes (B, E), and duration of the non-ventilatory period (C, F) in Testudines under hypoxic (A–C) and hypercarbic (D–F) exposures. *Chelonia mydas*: ◊ 25 °C, *Jackson, Kraus & Prange, 1979*; *Chelonoidis carbonarius*: ★ 25 °C, this study; *Chelydra serpentina*: ◈ 25 °C, *Boyer, 1966*; ◈ 22–24 °C, *West, Smits & Burggren, 1989*; ♣ 20 °C, *Frische, Fago & Altimiras, 2000*); *Chrysemys picta*: ◓ 20–23 °C, *Milsom & Jones, 1980*; ◑ 20 °C, *Glass, Boutilier & Heisler, 1983*; ◐ 30 °C, *Glass, Boutilier & Heisler, 1983*; ◯ 20 °C, *Silver & Jackson, 1985*; ⊗ 22–23 °C, *Milsom & Chan, 1986*; ⊙ 20 °C, *Funk & Milsom, 1987*; ◯ 30 °C, *Funk & Milsom, 1987*; *Gopherus polyphemus*: ◁ 22 °C, *Ultsch & Anderson, 1988*; *Pelomedusa subrufa*: ♠ 25 °C, *Burggren, Glass & Johansen, 1977*; ♥ 25 °C, *Glass, Burggren & Johansen, 1978*); *Phrynops geoffroanus*: ◆ 25 °C, *Cordeiro, Abe & Klein, 2016*; *Podocnemis unifilis*: ☐ 25 °C, *Cordeiro, Abe & Klein, 2016*; *Terrapene carolina* : ✳ 20–23 °C, *Altland & Parker, 1955*; ▷ 22 °C, *Ultsch & Anderson, 1988*; *Testudo horsfieldi*: ♰ 25 °C, *Benchetrit, Armand & Dejours, 1977*; ♀ 30 °C, *Benchetrit, Armand & Dejours, 1977*; ♯ 23–25 °C, *Benchetrit & Dejours, 1980*; *Testudo pardalis*: ✕ 25 °C, (continued on next page…)

**Figure 7 (…continued)**
*Burggren, Glass & Johansen, 1977*; ┼ 25 °C, *Glass, Burggren & Johansen, 1978*; *Trachemys scripta*: ▲
24 °C, *Jackson & Schmidt-Nielsen, 1966*; ▽ 28 °C, *Frankel et al., 1969*; ◣ 20 °C, *Jackson, 1973*; ◥ 30 °C,
*Jackson, 1973*; ▼ 20 °C, *Jackson, Palmer & Meadow, 1974*; ◭ 30 °C, *Jackson, Palmer & Meadow, 1974*; ◁
20 °C, *Hitzig & Nattie, 1982*; ▼ 25 °C, *Hicks & Wang, 1999*; ◀ 27–28 °C, *Johnson & Creighton, 2005*; ◮
20–23.5 °C, *Lee & Milsom, 2016*; △ 25 °C, this study.

Comparing our data with previous studies on the effect of hypoxia or hypercarbia on
ventilation and gas exchange in *Trachemys scripta*, *Frankel et al. (1969)* found values for
$T_{TOT}$ about three times larger during normoxia, hypoxia and hypercarbia when compared
to our study. However animals in their study had their tracheas cannulated which may
have influenced the length of the ventilatory cycle, since $T_{TOT}$ values reported by *Vitalis &
Milsom (1986b)* (calculated from their f': 1.7 s during normoxia and 4% $O_2$ and 1.8 s during
3–5% $CO_2$) are similar to ours. *Reyes & Milsom (2009)* report similar values for $f_E$ as in the
present study (from 8.4 ± 1.6 in normoxia during winter up to 37.1 ± 2.3 episodes.h$^{-1}$ in
summer), but found considerable variation in $f_{Repi}$ through different seasons, ranging from
3.6 ± 0.4 breaths.episode$^{-1}$ in normoxia during winter up to 26.1 ± 5.4 breaths.episode$^{-1}$
in hypoxic-hypercarbia during autumn, thereby demonstrating considerable seasonal
variation in breathing pattern in *T. scripta*. *Lee & Milsom (2016)* report nearly identical
values as in the present study for $f_{Repi}$ and $f_E$ during normoxia and hypoxia, and *Frankel
et al. (1969)* report a comparable $f_{Repi}$ during normoxia. *Johnson & Creighton (2005)*, on
the other hand, report greater values of $f_{Repi}$ during both normoxia and hypercarbia, and
*Frankel et al. (1969)* found $f_{Repi}$ at 10–12% $CO_2$ to be 5.6 ± 1.0 at 28 °C.

More data are available regarding $V_T$, $f_R$, $\dot{V}_E$, $\dot{V}O_2$, and $\dot{V}_E/\dot{V}O_2$ during both, hypoxic
and hypercarbic exposures. In general, data obtained in the present study for normoxia
are similar to the ones obtained by other authors, such as $\dot{V}O_2$, which at 25 °C varies
from 0.82 (*Hicks & Wang, 1999*; this study) to 1.1 mlO$_2$ kg$^{-1}$ min$^{-1}$ (24 °C; *Jackson &
Schmidt-Nielsen, 1966*), whereas the values given by *Vitalis & Milsom (1986b)* for $V_T$ and
$\dot{V}_E$ are the lowest ones reported for *T. scripta* exposed to hypoxia or hypercarbia. The overall
changes observed in the ventilatory responses of *T. scripta* to hypoxia and hypercarbia are
also comparable between the present study and data from the literature. Only $\dot{V}_E/\dot{V}O_2$ in
the present study, both during hypoxia and hypercarbia, was greater when compared to
data from the literature. This difference was caused by a much lower $\dot{V}O_2$ during hypoxic
and hypercarbic exposures when compared to data from other authors, since $\dot{V}_E$ was very
similar to data obtained by others at similar temperatures (*Jackson, Palmer & Meadow,
1974*; *Lee & Milsom, 2016*). The oxygen consumption measured by us during hypercarbia
was similar to the one obtained by *Jackson, Palmer & Meadow (1974)* at 10 °C, a 15 °C
difference, that may represent a variation in chemosensity seen in this species during
different seasons (*Reyes & Milsom, 2009*), as we found similarly low $\dot{V}O_2$ values during
hypoxic exposures. Interestingly, our normoxic $\dot{V}O_2$ values were well within the range for *T.
scripta* at 25 °C reported in the literature (*Hicks & Wang, 1999*; *Jackson & Schmidt-Nielsen,
1966*). A significant drop in oxygen consumption during hypoxia has also been described
before (*Jackson & Schmidt-Nielsen, 1966*; *Jackson, 1973*; *Lee & Milsom, 2016*), whereas other
studies did not find a pronounced fall in metabolism during hypercarbia (*Hicks & Wang,*

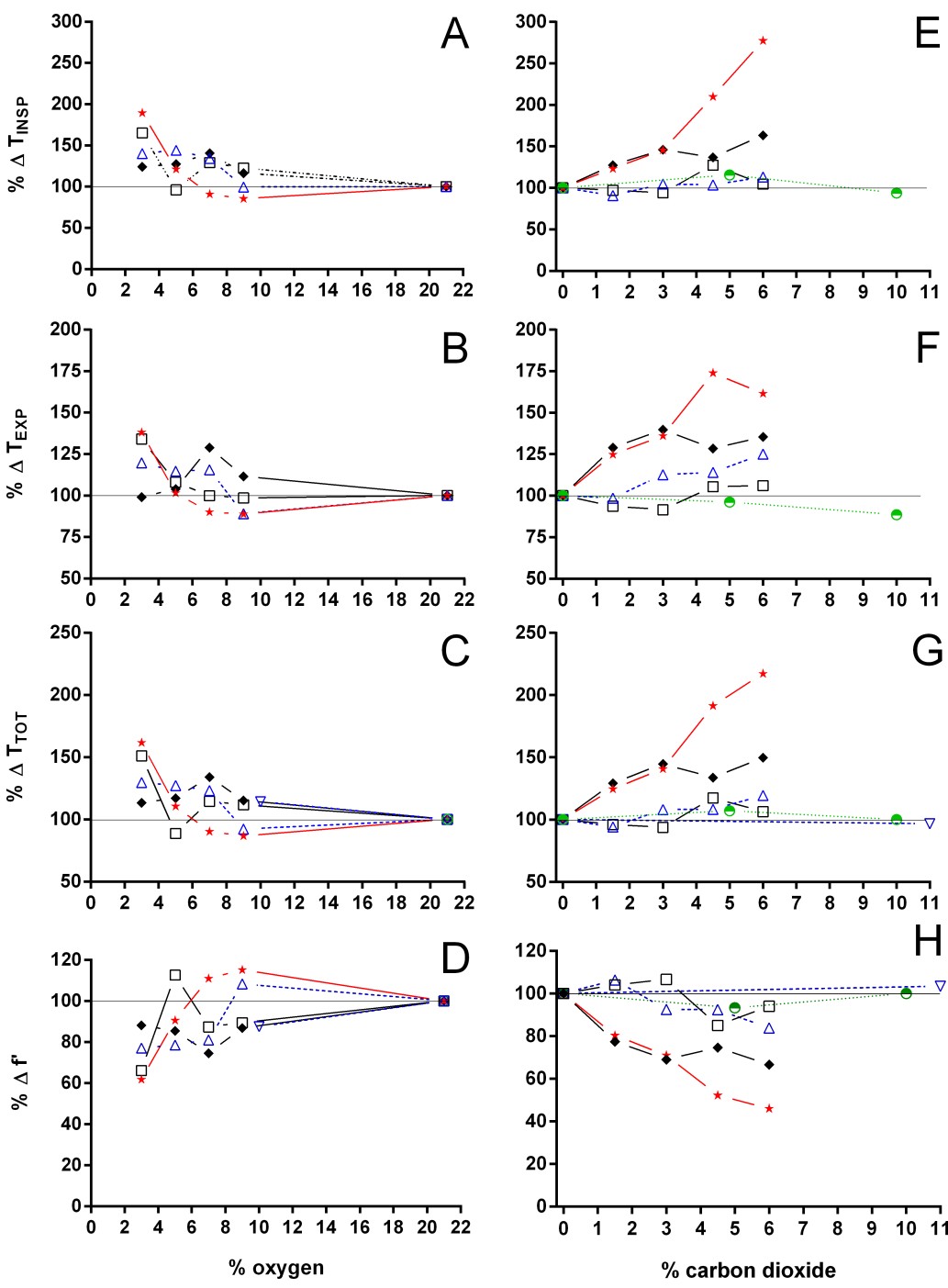

**Figure 8** **Relative changes in duration of inspiration and expiration, total duration of one ventilatory cycle, and instantaneous breathing frequency in Testudines under hypoxic and hypercarbic exposures.** Relative changes in duration of inspiration (A, E), duration of expiration (B, F), total duration of one ventilatory cycle (C, G), and instantaneous breathing frequency (D, H) in Testudines under hypoxic (A–D) and hypercarbic (E–H) exposures. For symbols see Fig. 7.

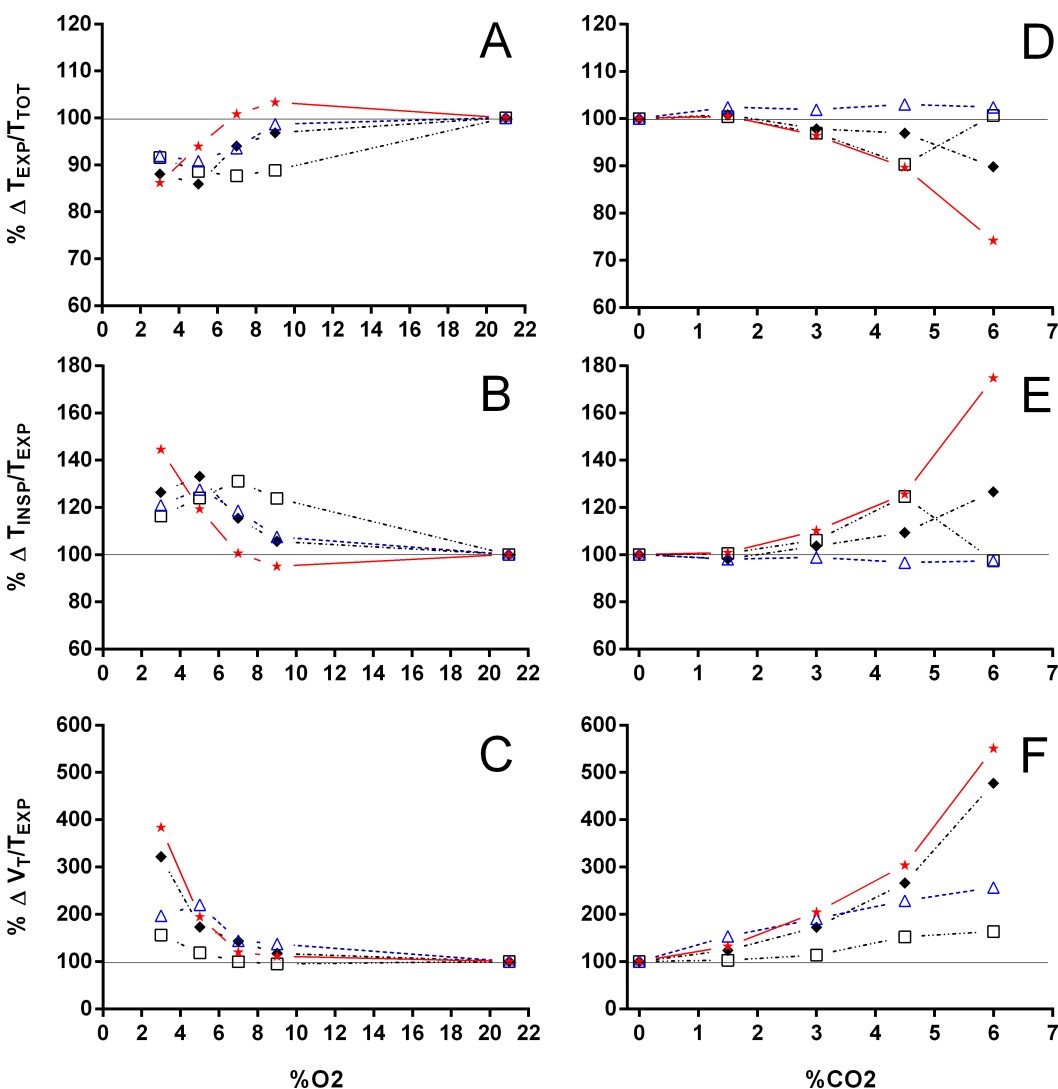

**Figure 9** **Relative changes in the relation between expiration and total duration of one ventilatory cycle, the relation between inspiration and expiration, and expiratory flow rate.** Relative changes in the relation between expiration and total duration of one ventilatory cycle (A, D), the relation between inspiration and expiration (B, E), and the expiratory flow rate (C, F) in Testudines under hypoxic (A–C) and hypercarbic (D–F) exposures. For symbols see Fig. 7.

*1999*; *Jackson, Palmer & Meadow, 1974*). One motive for the observed variations could lie in the significant seasonal variations in metabolism, gas exchange, and, consequently, ventilation found in *T. scripta* (*Reyes & Milsom, 2009*), variations that possibly were not eliminated by maintaining the animals at a constant temperature of 25 °C. Furthermore, exposing animals for 2 hours to each gas mixture may not have been sufficient to reach a physiological steady-state, as suggested by *Malte, Malte & Wang (2016)*. Another reason for this discrepancy could be the species physiological phenotypic plasticity, since animals used in the previous studies were native to the North American continent and thereby

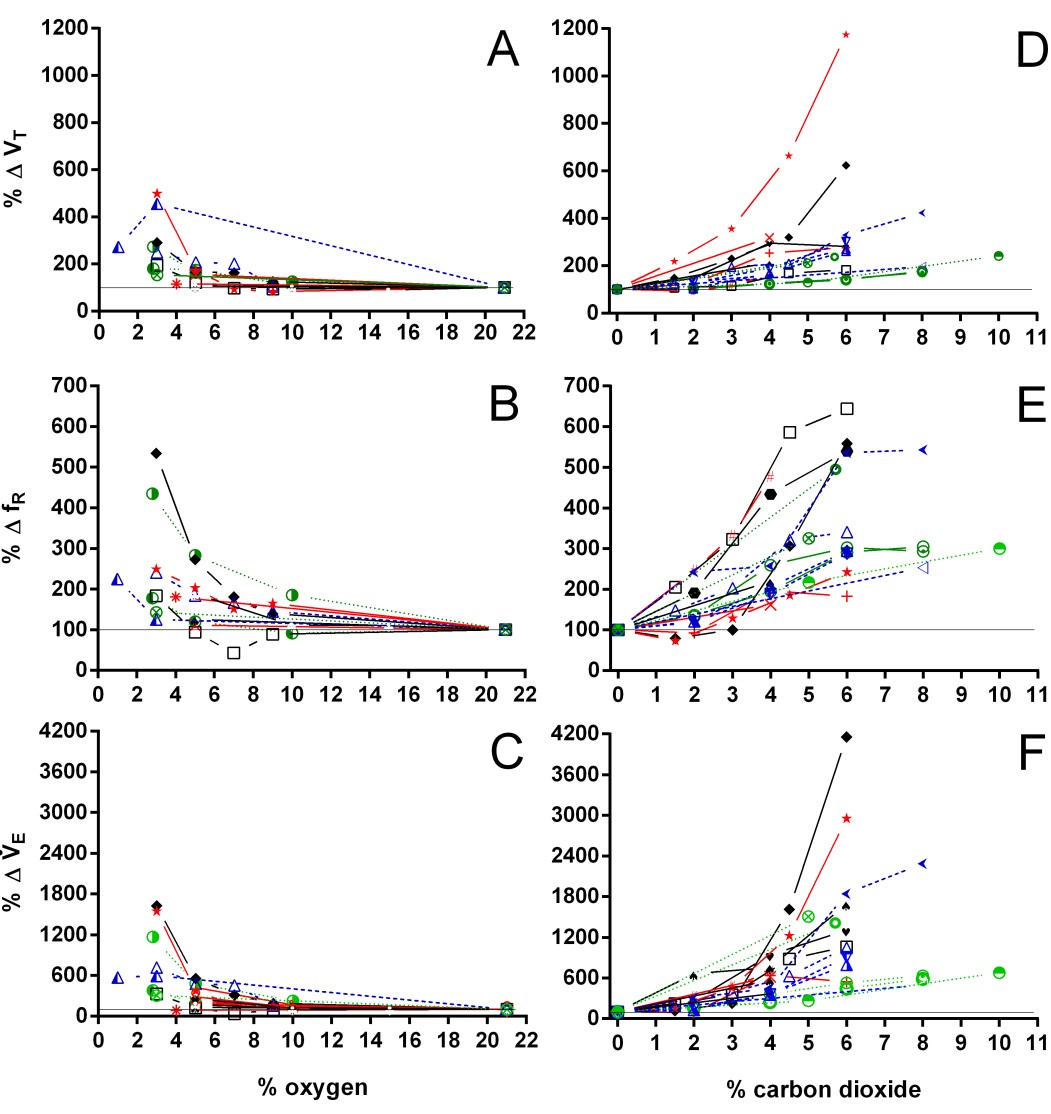

**Figure 10** Relative changes in tidal volume, breathing frequency, and minute ventilation in Testudines under hypoxic and hypercarbic exposures. Relative changes in tidal volume (A, D), breathing frequency (B, E), and minute ventilation (C, F) in Testudines under hypoxic (A–C) and hypercarbic (B–F) exposures. For symbols see Fig. 7.

subject to a more temperate climate than the animals used in the present study, that have been bred under the subtropical climate of southeastern Brazil.

The values for minute ventilation in *T. scripta* at 8% $CO_2$ found by *Hitzig & Nattie (1982)* seem somewhat low, when compared to the values found by *Johnson & Creighton (2005)* at the same $CO_2$ concentration at a different temperature (20 versus 27–28 °C, respectively), but are somewhat similar to the values found by *Jackson, Palmer & Meadow (1974)* at 20 °C and 6% $CO_2$ (135.0 versus 215 ml kg$^{-1}$ min$^{-1}$, respectively). The general response of *T. scripta* to reducing oxygen concentrations can be described by a moderate, when compared to the response during hypercarbia, increase in minute ventilation, mainly

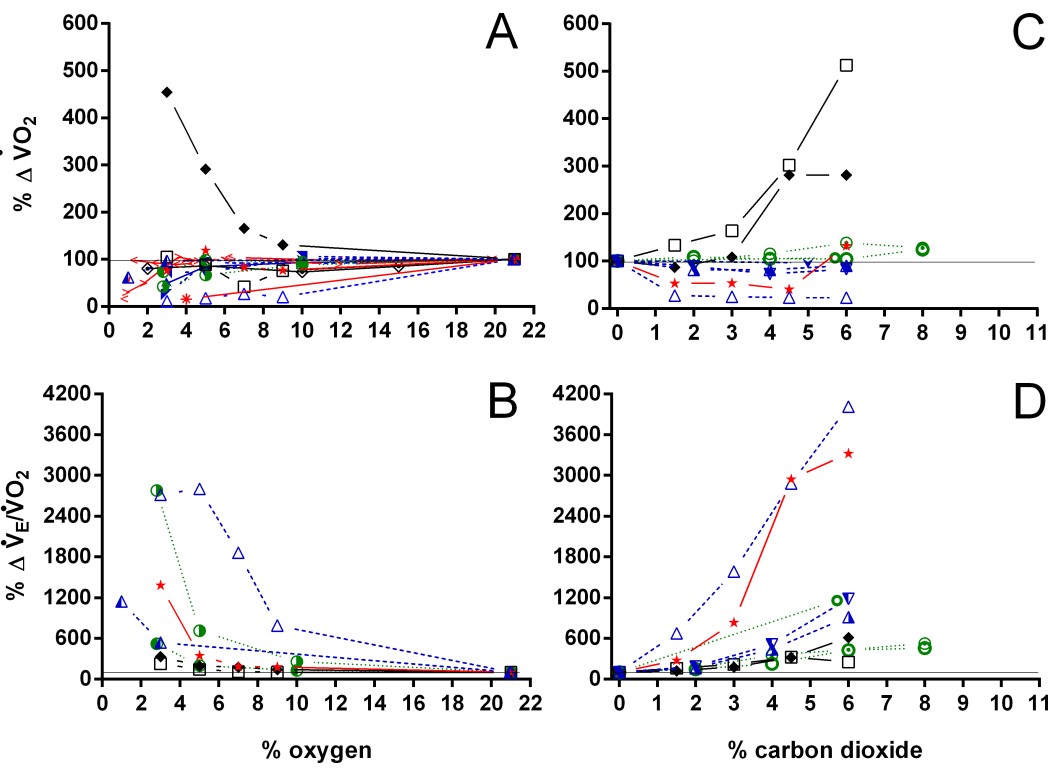

**Figure 11** **Relative changes in oxygen consumption and air convection requirement in Testudines under hypoxic and hypercarbic exposures.** Relative changes in oxygen consumption (A, C), and air convection requirement (B, D) in Testudines under hypoxic (A–B) and hypercarbic (C–D) exposures. For symbols see Fig. 7.

caused by increasing $V_T$, and a reduction in oxygen consumption, thereby increasing the air convection requirement. These changes are generally more pronounced below 5% $O_2$. The response to hypercarbia also includes an increase in ventilation due to an increase in $V_T$ and $f_R$. In *T. scripta* neither hypoxia nor hypercarbia caused significant changes in $T_{INSP}$, $T_{EXP}$, $T_{TOT}$, $f'$, and $f_{Repi}$, whereas $f_E$ and $T_{NVP}$, increased and decreased significantly, respectively.

In respect to *C. carbonarius* during hypoxic or hypercarbic exposures, only data on $V_T$, $f_R$, $\dot{V}_E$, and $\dot{V}O_2$ are available for other terrestrial Testudines belonging to the Emydidae and Testudinidae. Despite comparing different species, the ventilatory variables are similar among the terrestrial species studied, with the exception of the normoxic $\dot{V}O_2$ value given by *Altland & Parker (1955)* for *Terrapene carolina carolina*, possibly indicating that animals in their study may not have been resting quietly during normoxia. However, their $\dot{V}O_2$ value reported for 3–5% $O_2$ is identical to the values from other studies at similar oxygen concentrations. $V_T$ in *C. carbonarius* is on the lower end of data available for terrestrial Testudines, which may have been influenced by the relative large amount of bone tissue present in adult individuals of this species (AS Abe, pers. obs., 2005). Breathing frequency and $\dot{V}_E$, on the other hand, were very similar to the data obtained on other terrestrial Emydidae and Testudinidae (*Altland & Parker, 1955*, *Benchetrit, Armand*

*& Dejours, 1977*, *Burggren, Glass & Johansen, 1977*, *Glass, Burggren & Johansen, 1978* and *Benchetrit & Dejours, 1980*).

*Ultsch & Anderson (1988)*, studying *Gopherus polyphemus* and *Terrapene carolina*, found values of oxygen consumption very similar to those of *C. carbonarius* during both normoxia and hypoxia. Interestingly, *G. polyphemus* spends a significant amount of time in burrows that may show hypoxia as well as hypercarbia, and whose critical oxygen level (percentage of $O_2$ where $\dot{V}O_2$ starts decreasing) can be found at approximately 1.5% $O_2$, whereas the exclusively terrestrial *T. carolina* shows a somewhat larger critical oxygen tension of 3.5% $O_2$ (*Ultsch & Anderson, 1988*). Since *C. carbonarius* did not show any significant changes in $\dot{V}O_2$ during hypoxia down to 3% $O_2$, the critical oxygen level of this species seems to be similar to the one seen in the former two species, but $\dot{V}O_2$ was consistently lower at any oxygen concentration when compared to *G. polyphemus* and *T. carolina* and e.g., at 3% $O_2$ (0.08 $mlO_2$ $kg^{-1}$ $min^{-1}$) was similar to the lowest $\dot{V}O_2$ given for *G. polyphemus* (0.05 $mlO_2$ $kg^{-1}$ $min^{-1}$) and *T. carolina* (0.08 $mlO_2$ $kg^{-1}$ $min^{-1}$) at less than 1% $O_2$ (*Ultsch & Anderson, 1988*). *C. carbonarius* is not known to use burrows and therefore may not show a critical oxygen level as low as *G. polyphemus*, but *Chelonoidis chilensis* has been reported to use shallow burrows for retreat during cold days (*Pritchard, 1979*) and therefore other species of the Testudinidae may possess a similarly low oxygen level as the testudinidid *G. polyphemus*.

## Relative changes in respiratory variables

Analyzing the respiratory variables available in the literature for chelonians exposed to hypoxia and hypercarbia (Figs. 7–11), one notices the discrepancy in data availability between commonly studied parameters such as $V_T$, $f_R$, $\dot{V}_E$, and $\dot{V}O_2$, and less frequently reported ones such as $T_{EXP}$, $T_{TOT,}$ or $f_E$, for example. Furthermore, only very few terrestrial species have been studied, when compared to the wealth of data available for *T. scripta* and *C. picta*. Based on the data analyzed, it seems clear that the breathing pattern of terrestrial chelonians does not significantly differ from aquatic or semi-aquatic species when considering the responses to hypoxia and hypercarbia. With few exceptions, both hypoxia and hypercarbia elicit similar respiratory responses, showing variation mainly in the magnitude of the species' responses. The different patterns seen in $f_E$ and $f_{Repi}$ during hypoxia and hypercarbia may suggest varying degrees of chemosensivity between species and towards different gas exposures. Previous experimental manipulations transforming episodic breathing into continuous single ventilations in *T. scripta* were vagotomy (*Vitalis & Milsom, 1986b*) and dissection of the spinal cord (*Johnson & Creighton, 2005*). Recently, *Johnson, Krisp & Bartman (2015)* changed episodic breathing in *T. scripta* from episodic to singlet breathing through pharmacological manipulation of serotonin 5-HT$_3$ receptors. Studying the participation of serotonin in central chemoreception under hypoxia and hypercarbia in phylogenetically distant species, as well as species occupying different habitats, might help to elucidate the varying responses to hypoxia and hypercarbia seen in chelonian breathing pattern, since the switch from episodic to singlet breathing under hypoxia has been suggested to be caused by an increased respiratory drive (*Fong, Zimmer & Milsom, 2009*). However, under hypercarbia nearly all species increase the number of

breaths per episode, and do not decrease $f_{Repi}$ as under hypoxia, suggesting that $CO_2$ exposure increases respiratory drive by different regulatory pathways than under hypoxia. Interestingly, *Herman & Smatresk (1999)* demonstrated that in *T. scripta* hypoxia and hypercarbia cause different changes in pulmonary ventilation and perfusion. During hypoxia, lung ventilation and perfusion increased, whereas under hypercarbia only lung ventilation increased, but not pulmonary perfusion, resulting in a ventilation/perfusion mismatch during exposure to $CO_2$. *Burggren, Glass & Johansen (1977)* found a similar cardiovascular response in *Pelomedusa subrufa* and in *Testudo pardalis*, suggesting a common testudine response, but its importance or relation with the differences observed in $f_{Repi}$ remains unclear.

*P. geoffroanus* and *C. carbonarius* seem to be more sensitive regarding $T_{INSP}$, $T_{EXP}$, $T_{TOT}$, f', $T_{EXT}/T_{TOT}$, and $T_{INSP}/T_{EXP}$, with the former species increasing these variables mainly during hypercarbia, but the latter one increasing all variables with increasing levels of hypoxia and hypercarbia. Such increases in $T_{INSP}$ and $T_{EXP}$ have been interpreted by *Johnson, Krisp & Bartman (2015)* as a stronger respiratory drive from central respiratory neurons, whose intensity, however, seems to vary among species. The absolute and relative decrease in instantaneous breathing frequency seen in *C. carbonarius* implies that breathing mechanics may be more variable than previously anticipated for Testudines, since *Vitalis & Milsom (1986b)* found f' to be unaffected by either hypoxia or hypercarbia in *T. scripta* and suggested (*Vitalis & Milsom, 1986a*; *Vitalis & Milsom, 1986b*) that *T. scripta* breathes at combinations of volume and frequency to keep the mechanical work of breathing at a minimum. In the present study, f' in *T. scripta*, as well as in *C. carbonarius*, did show larger variations than reported for *T. scripta* in earlier studies (*Frankel et al., 1969*; *Vitalis & Milsom, 1986b*). *Vitalis & Milsom (1986a)* found, based on mechanical analyses of the respiratory system of *T. scripta*, that the mechanical work of breathing is minimal at ventilation frequencies of 35 to 45 cycles $min^{-1}$ for different levels of minute pump ventilation (100, 200, 300 ml $min^{-1}$), meaning that for a minute pump ventilation of 200 ml $min^{-1}$. Animals should therefore ventilate at a frequency of 40 breaths $min^{-1}$ and a tidal volume of 5 ml to ventilate the respiratory system with the lowest mechanical work, but such a breathing pattern would result in severe alkalosis due to increased $CO_2$ excretion (*Vitalis & Milsom, 1986b*). In the present study, however, *T. scripta* reached the greatest level of minute ventilation (215.9 ml $min^{-1}$ $kg^{-1}$) at 6% $CO_2$, using a tidal volume of 57.2 ml $kg^{-1}$ and an instantaneous breathing frequency of 18.3 breaths $min^{-1}$ ($f_R = 3.0$ breaths $min^{-1}$), values much different from mechanical predictions. The significance of this variation in breathing pattern versus the mechanical predictions of work of breathing needs to be investigated to better understand the mechanical work of breathing of the Testudines respiratory system, since mechanical work of breathing increases markedly with increasing tidal volume, e.g., from 57 to 272 ml $cmH_2O$ $min^{-1}$ $kg^{-1}$ at 6.2 ml $kg^{-1}$ and 3.0 breaths $min^{-1}$ in undisturbed *T. scripta* versus 34.2 and 0.8 breaths $min^{-1}$ in vagotomized *T. scripta*, each at 4% $O_2$ (*Vitalis & Milsom, 1986b*).

The relatively large increases seen in $V_T/T_{EXP}$ of *C. carbonarius* and *P. geoffroanus* can be explained by very low values of $V_T$ under normoxic conditions. *P. geoffroanus* (3.1 ml $kg^{-1}$; *Cordeiro, Abe & Klein, 2016*) and *C. carbonarius* (3.98 ml $kg^{-1}$; this study) show much

smaller tidal volumes during normoxia than other chelonians (mostly between 10 and 20 ml kg$^{-1}$), resulting in relatively larger increases in $V_T$ during hypoxia and hypercarbia than the other species. Both species also showed relatively larger increases in $\dot{V}_E$, which are again attributable to the low values seen in $f_R$ and $V_T$ under normoxic conditions.

Whereas $\dot{V}_E$ increases largely in all species, $\dot{V}O_2$ remains unaltered or even decreases under both hypoxia and hypercarbia in nearly all species investigated. The relative increase seen in *P. geoffroanus* under both hypoxia and hypercarbia can be explained by the very low oxygen consumption under normoxic conditions, which could be a consequence of significant extra-pulmonary gas exchange or hypometabolism in this species (*Cordeiro, Abe & Klein, 2016*). Increases in $\dot{V}_E/\dot{V}O_2$ have been linked both under hypoxia (e.g., *Glass, Boutilier & Heisler, 1983*) and hypercarbia (e.g., *Funk & Milsom, 1987*) to regulation of arterial $PO_2$, $PCO_2$, and pH, as all turtles investigated maintain control of these variables under varying environmental conditions.

The chelonian respiratory system shows significant variations in lung structure, as well as in associated structures such as the post-pulmonary septum (PPS; *Perry, 1998*). The PPS is a membrane that partially or completely separates the lungs from the other viscera (*Lambertz, Böhme & Perry, 2010*). As a testudinid, *C. carbonarius* possesses a complete post-pulmonary septum (W Klein, pers. obs., 2015), when compared to the incomplete PPS of the emydid *T. scripta* (*Lambertz, Böhme & Perry, 2010*). The presence or absence of a PPS may significantly influence the mechanics of the respiratory system, as has been shown for the post-hepatic septum of the lizard *Salvator* (*Tupinambis*) *merianae*, whose static breathing mechanics was significantly affected by the removal of their post-hepatic septum (*Klein, Abe & Perry, 2003*). Similarly, a complete PPS in Testudinidae could alter the mechanics of the respiratory system by reducing the impact of the viscera onto the lungs, when compared to species with an incomplete PPS such as *T. scripta*.

## CONCLUSION

This is the first study to present all the different variables necessary to fully characterize the breathing pattern in the terrestrial *C. carbonarius* and the semi-aquatic *T. scripta* during hypoxic and hypercarbic conditions. Contrary to most previous reports on breathing pattern in terrestrial Testudines, *C. carbonarius* did show considerable non-ventilatory periods with more than one breath per episode. While our data confirm previous data on the general response of *T. scripta* to hypoxia and hypercarbia, breathing pattern has been found to diverge significantly from predictions based on mechanical analyses of the respiratory system.

Our meta-analysis demonstrates general trends regarding ventilatory parameters of Testudines when exposed to hypoxia or hypercarbia, but a multivariate analysis of the taxons respiratory physiology will need a complete set of ventilatory parameters from a much larger number of species. To date it is not possible to associate the variations in the magnitude of different respiratory variables with phylogeny, habitat, behavior, and/or lung structure, which could provide important information regarding the evolution of cardiorespiratory physiology in chelonians. Cardiovascular data regarding intracardiac

shunt, pulmonary and systemic perfusion, and blood gases during hypoxia and hypercarbia are particularly needed from more species to fully understand blood gas homeostasis in such an important group of intermittent breathers.

### Funding

Financial support was provided by the Fundação de Amparo à Pesquisa do Estado de São Paulo (FAPESP) to Wilfried Klein (2012/18652-1) and Augusto Shinya Abe and Wilfried Klein (2008/57712-4), by the Conselho Nacional de Desenvolvimento Tecnológico e Científico (CNPq) to Augusto Shinya Abe and Wilfried Klein (573921/2008-3). Pedro Trevizan-Baú received a scholarship from the Coordenação de Aperfeiçoamento de Pessoal de Nível Superior (CAPES). The funders had no role in study design, data collection and analysis, decision to publish, or preparation of the manuscript.

### Grant Disclosures

The following grant information was disclosed by the authors:
Fundação de Amparo à Pesquisa do Estado de São Paulo (FAPESP): 2012/18652-1, 2008/57712-4.
Conselho Nacional de Desenvolvimento Tecnológico e Científico (CNPq): 573921/2008-3.
Coordenação de Aperfeiçoamento de Pessoal de Nível Superior (CAPES).

### Competing Interests

The authors declare there are no competing interests.

### Author Contributions

- Pedro Trevizan-Baú conceived and designed the experiments, performed the experiments, analyzed the data, prepared figures and/or tables, authored or reviewed drafts of the paper, approved the final draft.
- Augusto S. Abe conceived and designed the experiments, contributed reagents/materials/analysis tools, authored or reviewed drafts of the paper, approved the final draft.
- Wilfried Klein conceived and designed the experiments, performed the experiments, analyzed the data, contributed reagents/materials/analysis tools, prepared figures and/or tables, authored or reviewed drafts of the paper, approved the final draft.

### Animal Ethics

The following information was supplied relating to ethical approvals (i.e., approving body and any reference numbers):

Experiments were performed between November 2014 and February 2015 following approval by the Instituto Chico Mendes de Conservação da Biodiversidade (SISBIO; license number 35221-1) and Comissão de Ética no Uso de Animais (CEUA USP/Campus de Ribeirão Preto; protocol number 12.1.1541.53.0).

## Data Availability

Klein, Wilfried (2018): *C. carbonarius* and *T. scripta* ventilation. figshare. Fileset. Available at https://doi.org/10.6084/m9.figshare.5936296.v1.

## Supplemental Information

Supplemental information for this article can be found online at http://dx.doi.org/10.7717/peerj.5137#supplemental-information.

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
