# Peer review of "Effects of environmental hypoxia and hypercarbia on ventilation and gas exchange in Testudines"

_PeerJ, doi:10.7717/peerj.5137_

## Round 0.1 · original submission · Minor Revisions

Overview
Both reviewers found that your study made a useful contribution, both in its new empirical data and its comparative/meta-analysis. However, they did indicate that minor revision was required. Reviewer 1 indicated a need for additional detail on the methods and some of the results (Fig. 1), a potential alternative interpretation and improvements in the organization and grammar. Reviewer 2 made quite a few suggestions for improvements in the use of English directly on the manuscript. Included in these comments were a few more substantial comments.

To facilitate your revision, I have considered the comments on word use and grammar from both reviewers and incorporated them into a pdf with my own suggestions. Therefore, you can use my annotated pdf in correcting the language for your revision. In that pdf, I highlighted words of concern and used inserted comments to suggest an alternative or explain the problem or both.

I have also repeated the more substantial suggestions in the annotated pdf of Reviewer 2 after my comments below to make it easier to incorporate them into your ‘rebuttal’.

Major concerns

I think that you assume too much understanding of turtle phylogeny by readers. Several times in the manuscript, you refer to sub-orders or families for which the general reader has no context. Consider starting the introduction with a brief overview of the major groups of turtles and the varied demands on their breathing systems. This could lead naturally into a discussion of the relatively small and uneven distribution of species that have been studied and then to the limited number of respiratory variables measured even in the studied species. Later, when first mentioning a species, you can make it clear which family and broader group they represent. Adding family names to Table S1 would aid your presentation and readers as well.

L164-182. Given the importance given to the literature review in text and figures, your comparative analysis method should have a should have a separate paragraph in the objectives. Did you expect to see differences among species? If so, what were they? Even though your study is not formally a meta-analysis, it would be appropriate to provide information similar what is required in formal meta-analyses. You should indicate how you selected the studies to include, how you searched for relevant studies, whether any were excluded, whether the included studies form a complete set of studies meeting your criteria or simply a sub-set that happened to be familiar or available. The long list of references is rather awkward in the text. Furthermore, from your statements, I infer that not all studies provide data for all panels and figures. Would it be possible to link the studies included in the comparative analysis to Table S1 to see the range of species studied? Perhaps a more complete table indicating which variables as well as which species would be useful and along with a list of figures and panels in which the study was used. This may be more appropriate in to include in the main article, rather than supplementary material.

L139ff. The methods on experimental design are incomplete. They indicate that after normoxic data were obtained, a sequence of 4 treatment levels of either hypoxic or hypercarbic conditions at 2-h intervals was provided. However, they don’t mention when the alternative treatment was provided, if it was at all. This has important implications for sample size and could potentially affect interpretation. Furthermore, the Results refer to returning to normoxia, but this is not mentioned in the Methods.

L190. You refer to interspecific differences here. However, an interspecific comparison was not one of your explicit objectives and it is not clear whether a two-species comparison should be a logical component of this study. Furthermore, data to support the interspecific differences seem to be missing. Please either incorporate the logic of comparing species into the objectives and provide the relevant data, perhaps in a table, or remove the comparison.

L203-204. The Methods and Results give expiratory flow rate as T_T/T_EXP whereas Fig. 5 gives the reverse ratio. This is very confusing and might be serious if the error is in calculation for the figure rather than labeling the axis. Please meticulously check all your values and symbols for consistency and reconfirm the accuracy of all calculations and descriptions. For some of the other abbreviations, you switch between capital and lower case letters in the captions and axis labels.

Inconsistent order of your species and variables makes the manuscript unnecessarily hard to follow. Please decide on the order with which you will present the two species in your results and use the same order throughout the manuscript, including the abstract, methods, figures and figure captions and discussion.

The same logic applies to the variables. For example, it is hard to verify the statement on L190 because the order does not match the figures. Indeed, the order of variables changes between methods, results and figures. The inconsistency between text and figures will annoy readers if they have to jump back and forth among figures as they are reading. Please decide on a logical grouping and order for the variables and present them consistently in that order. Use these logical groupings for the figures, and don’t allocate variables between figures based simply on a the number of panels. The figures for the experimental data and comparative data should have the same groupings. Finally, the variables should be discussed in the same order, although I understand that there may be some situations where some variables need to be explained in relation to other variables not in the same sequence.

I can understand the dilemma that lead you to produce a combined Results and Discussion section because the comparative findings are to some extent results. Nevertheless, in its present form the Discussion is very long, hard to follow and may discourage readers. My suggestion is that you have a section called ‘Experimental Results’ which is dedicated only to your experimental results. This would be followed by a ‘Comparative Results and Discussion’ section in which you would present the data on other species along with an integrated discussion of your results and the literature findings. This will reduce some overlap. This section should have sub-headings to allow the reader to follow easily. Ideally these sub-headings would correspond to a logical organization of the variables as requested above. Furthermore, check each paragraph to be sure that you have a strong topic sentence to guide the reader and a final sentence that pulls together the main conclusion from that paragraph. Finally, add a ‘General Discussion’ section in which you highlight the broader findings and future research needs and eliminate the ‘Conclusions’ section.

Wouldn’t it be logical to include intermittent breathing in your comparative analysis?

Several times you argue that the set of variables studied in most of the literature is incomplete. However, I did not feel you made the case clearly for why this is a problem. Do you just want completeness for its own sake or are there important conceptual advantages to doing so? How might we fail to understand turtle breathing if we are missing certain variables?

Similarly, try to make the case more explicitly for your conclusion that so far there is no correlation between turtle breathing patterns and phylogeny, habitat, or morphology.

Lesser concerns

L34-37. Please revise sentence. The part of this sentence about the tortoise is redundant and the part about the turtle is vague. Try to focus on broader implications in relation to the literature. For example, ‘Our study shows that terrestrial Testudines sometimes use episodic breathing, previously attributed mainly to aquatic turtles.’

L54, 152. You use the term ‘number of breathing episodes’ for f_Repi and ‘overall breathing frequency’ for f_R. I would think that both are frequencies or rate since number of episodes makes no sense without the duration of observation (and indeed the units on the figures bear this out). Unless this is established terminology in the field, please change to frequency throughout.

L67-71. This sentence is confusing. You just introduced lack of data on terrestrial Testudinae and then start discussing a semi-aquatic species in a different family. Reverse the sentence to emphasize the tortoise pattern and then mention the contrast with semi-aquatic species. Follow this with the sentence supporting the pattern in other species (L74-79) and then the more general hypothesis that the contrast in ventilator patterns may be related to the costs of surfacing in aquatic species (L71-74). However, I would not focus on this as only an adaptive response for a particular aquatic species but as a possible adaptation for the contrasting environments in for all species.

L78. The reference to effect of hypoxia is confusing here because it changes the focus from the point of the paragraph. Leave for the next section.

L79. Having worked on costs of breathing for fishes, I wonder whether there are data showing that semi-aquatic species change their ventilation patterns when in and out of water (e.g., basking). I would think this would be relevant to the background argument.

L86. I don’t think readers would be expected to know what pleurodirans are. Instead of this, indicate the taxonomic relationship to the previously mentioned species, if relevant, or simply indicate the family.

L95. Is the reference to cryptodirans important here? If so, the two sub-orders should be introduced earlier when you are discussing the taxonomic diversity of studied species.

L151-160. Some abbreviations previously defined are repeated here. Either remove the abbreviations from the Introduction and provide all abbreviations here or use the abbreviations previously introduced. My opinion is that the Introduction might be easier reading for some readers if you used the full words there and introduced all abbreviations in this section, but it is your choice. Also, you should provide the units for each of your measures. Were any measures scaled to body size? If so, specify.

L193-194. This is a very confusing sentence. It needs to be rewritten in a more logical form. Furthermore, I cannot see symbols in Fig. 3 that support the statement about significant differences.

L276. Do you have a reference for seasonal effects on animals held in laboratories under constant conditions (even if you need to refer to species other than turtles)?

L328-337. This paragraph seems more appropriate to a Conclusions or General Discussion section, after you give the detailed discussion that supports it. You could indicate the types of species and variables needed and what that would provide for scientific understanding.

L345-355. Are these details of mechanistic interventions really relevant to this article? Their contribution is not clear as there is no conclusion from the summary of detailed studies.

L382-388. The paragraph on the septum seems out of place or extraneous. There is no clear link to the findings of this study. [I note, however, that Reviewer 2 found it interesting, so it may just need to have its relevance more clearly stated.]

L555. ‘after exposure’ is ambiguous because it could refer to an hour after the start of exposure to hypoxia/hypercarbia or an hour after the hypoxic/hypercarbic conditions were returned to normoxia. Clarify in all relevant captions.

I found that you often missed commas between independent clauses and sometimes inserted them where they were inappropriate. Please check the rules so that any new text will not have these errors.

Also check that tenses are used consistently throughout your results and discussion sections (past to describe the experimental results in this and previous studies and present to describe the current understanding).

Reviewer 2 comments from annotated pdf

L148. Probably not enough to establish a new steady state. See Malte et al J Exp Biol 219, 3810-21 (2016).

L345-346. Unlikely that the ‘mechanisms’ differ between species. [DLK: because this is one of your broader conclusions, consider this challenge seriously to see if your conclusion is valid and whether it needs additional supporting arguments.]

L354-355. Unclear!

L371. But this would lead to severe alkalosis.

L381. But is it important?

L382-388. This is interesting and could be abbreviated.

Reviewer 1 ·

Basic reporting

1.) Overall, the study was well framed and the data/meta-analysis make a nice contribution. However, I suggest the authors thoroughly proofread the paper for proper English grammar and syntax. For example, lines 82-83 read, “It has been shown, that the normal response to either of the changes…” Lines 97-98 read, “C. carbonarius was chosen because it is a wide-spread South American tortoise that not had its respiratory physiology investigated previously…” These kinds of errors were prevalent throughout the paper.
2.) Additionally, my view is that the paper would read more clearly if it included distinct “Results” and “Discussion” sections. As written, the discussion/interpretation is intertwined with the results. This makes assimilating trends observed in a large number of respiratory variables across multiple species rather difficult.
3.) It would be helpful if Figure 1 was, or included, raw data showing a “zoomed in” series of breaths (and also breath-to-breath O2 consumption) with all of the respiratory pattern parameters assessed in this study clearly labeled. This will improve the reader’s ability to follow along during the presentation of the results.
4.) In text citations with more than two authors were inconsistent. Some citations were reported as “Author A et al., date” while others were reported as “Author A, Author B, and Author C, date.” Please choose one style and adhere to that format.

Experimental design

1.) Although references to methods are cited, please include more methodological detail specific to the experiments performed for ventilation and O2 consumption measurements. (1) What was the approximate dead space volume in the mask and funnel and what were the flow rates for each experimental set up? Since the mask and funnel were, presumably, small volumes, flow rates need to be sufficiently high to prevent rebreathing. If flow rates were not high enough, rebreathing of the gases in the mask or funnel could alter interpretation of the present results. (2) How was the calibration and analysis for O2 consumption performed? Based on the references, it seems that the O2 fraction measured for each breath was determined from a calibration curve with known gas concentrations injected into the mask. Was this the case? A bit more detail would improve the ability to understand the experiments performed. (3) For calibration of the pneumotach, is the volume obtained from integrating the flow signal linearly across the physiologically relevant range of flow rates described in the results (~5-60 ml s-1; Fig. 5)? Clarification of each of these points in the text of the manuscript would improve my confidence in the data presented.
2.) Please clarify the numbers of animals used in each analysis. The authors indicate that 8 T. scripta and 6 C. carbonarius were used in this study. However, lines 144-147 read, “Experimentation started around 8:00 am and ventilation and gas exchange were measured under normoxic conditions, followed by progressively decreasing hypoxic (9, 7, 5, 3% O2) or progressively increasing hypercarbic (1.5, 3.0, 4.5, 6.0% CO2) exposures.” Was each animal exposed to both progressive hypercarbia and hypoxia, or were only some of the animals from each species exposed to either hypercarbia or hypoxia? If, say, half of the animals from each species were exposed to either hypoxia or hypercarbia, the “n” of each experiment is likely to be very low. Please clarify these points in the text of the manuscript and mark the “n” for each analysis in figure legend.

Validity of the findings

1.) With respect to interpreting the results from hypercarbia/hypoxia from the experiments and the meta-analysis, it is worth mentioning that intracardiac shunt patterns in reptiles can have unpredictable effects on blood gases during hypoxia and hypercarbia, Lines 345-346 state that variation in ventilatory sensitivity to gases observed among species may “suggest varying regulatory mechanisms of breathing pattern during hypoxia and hypercarbia and between species.” Although this may be true, it is also possible that mechanisms of breathing control are similar across species, but differences in intracardiac shunt patterns produce arterial gases that lead to differences in stimulatory signals at chemoreceptor sites during hypoxia/hypercarbia and therefore contribute to the range of ventilatory responses across species. For example, at least in toads, the right-left cardiac shunt decreases during hypoxia (Gamperl et al., 1999, J. Exp Biol. 202, 3647–3658). Thus Po2 and the magnitude of the ventilatory response to hypoxia, and hypothetically hypercarbia, is likely to be influenced by ventilatory “chemosensitivity” and cardiac shunt patterns.

·

Basic reporting

This is a very detailed descriptive study on the effects of hypercapnia and hypoxia on the ventilatory pattern in two species of chelonians. The study provides novel data on a hitherto un-described tortoise (Chelonoidis carbonarius) and make comparison to the much more well-characterized Trachemys scripta (the “white rat of reptile physiology”). The study shows that the differences in breathing pattern amongst terrestrial and aquatic testudines is not as marked as previously reported and this conclusion is of interest. Secondly, the manuscript provides an excellent review of the available data on the ventilatory responses to CO2 and hypoxia in turtles. This is a very useful resource for future studies.

I have made some minor suggestion in the annotated PDF that the authors may consider when revising the manuscript. Some of these are merely suggestions to improve wording or clarity; there are also other types of suggestions.
My only main concern is the very descriptive nature of the study and the manner in which the manuscript is written. Is would be desired if the authors could provide a more defined rationale for the study – what are the specific goals. Then, the discussion should follow up on these problems and make more firm statements based on the extensive literature review.

Tobias Wang

Experimental design

well performed

Validity of the findings

no concerns

Additional comments

This is a very detailed descriptive study on the effects of hypercapnia and hypoxia on the ventilatory pattern in two species of chelonians. The study provides novel data on a hitherto un-described tortoise (Chelonoidis carbonarius) and make comparison to the much more well-characterized Trachemys scripta (the “white rat of reptile physiology”). The study shows that the differences in breathing pattern amongst terrestrial and aquatic testudines is not as marked as previously reported and this conclusion is of interest. Secondly, the manuscript provides an excellent review of the available data on the ventilatory responses to CO2 and hypoxia in turtles. This is a very useful resource for future studies.

I have made some minor suggestion in the annotated PDF that the authors may consider when revising the manuscript. Some of these are merely suggestions to improve wording or clarity; there are also other types of suggestions.
My only main concern is the very descriptive nature of the study and the manner in which the manuscript is written. Is would be desired if the authors could provide a more defined rationale for the study – what are the specific goals. Then, the discussion should follow up on these problems and make more firm statements based on the extensive literature review.

Tobias Wang

---

## Round 0.2 · accepted · Accept

Thank you for your careful revision and detailed response to the comments. The manuscript is now suitable for publication. I have attached a pdf with a few minor corrections that can be addressed during production. It is also important that Fig. 7 - 11 have the symbols and sources presented in the first caption presenting the comparative material (Fig. 7) rather than supplementary data. I will indicate to PeerJ staff that they should provide a solution to the formatting problem.

#